# Mid-Atlantic U.S. observations of radiocarbon in $CO_2$: fossil and biogenic source partitioning and model evaluation

Bianca C. Baier[1], John B. Miller[1], Colm Sweeney[1], Scott Lehman[2], Chad Wolak[2], Joshua P. DiGangi[3], Yonghoon Choi[3,4], Kenneth Davis[5,6], Sha Feng[5,+], Thomas Lauvaux[5,*]

[1]NOAA Global Monitoring Laboratory, Boulder, CO 80305
[2]Institute of Arctic and Alpine Research, University of Colorado-Boulder, Boulder, CO 80309
[3]NASA Langley Research Center, Hampton, VA 23681
[4]Analytical Mechanics Associated, Hampton, VA 23666
[5]Department of Meteorology and Atmospheric Science, The Pennsylvania State University, University Park, PA 16802
[6]Earth and Environmental Systems Institute, The Pennsylvania State University, University Park, PA 16802
[+]Now at: Atmospheric, Climate and Earth Sciences Division, Pacific Northwest National Laboratory, Richland, WA 99352
[*]Now at: Groupe de Spectrométrie Moléculaire et Atmosphérique (GSMA), Université de Reims-Champagne Ardenne, UMR CNRS 7331, Reims, France 51100

*Correspondence to*: Bianca Baier (bianca.baier@noaa.gov)

**Abstract.** Accurately quantifying regional anthropogenic $CO_2$ fluxes is fundamental to improving our understanding of the carbon cycle and for creating effective carbon mitigation policies, and the radiocarbon to total carbon ratio in atmospheric $CO_2$ ($\Delta^{14}CO_2$) is a robust tracer of fossil fuel $CO_2$ that can discriminate between biogenic and fossil fuel $CO_2$ sources. NASA's Atmospheric Carbon and Transport-America (ACT-America) airborne mission between 2016 and 2019 aimed to improve the accuracy of regional greenhouse gas flux estimates, through refining understanding and characterization of fluxes and flux uncertainties in models. $\Delta^{14}CO_2$ observations from 26 flights are presented for examining seasonal $CO_2$ source partitioning in the Mid-Atlantic U.S. Observed variability in boundary layer $CO_2$ at time scales ranging from intra-day to seasonal was largely driven by biogenic $CO_2$ ($CO_{2bio}$) variability that ranged from -19.7 ppm in summer to 16.2 ppm in fall, while fossil fuel $CO_2$ ($CO_{2ff}$) variability remained at $3.3 \pm 2.0$ ppm. Carbonyl sulfide uptake was well-correlated with $CO_{2bio}$ uptake, and examining this relationship, and that between $CO_2$ and $CO_{2bio}$ variability reinforces the seasonal extent of gross primary productivity response throughout ACT-America. We use airborne $\Delta^{14}CO_2$ flask sampling alongside in situ carbon monoxide measurements to calculate high-frequency $CO_{2ff}$ and evaluate the magnitude and diurnal variability of modeled $CO_{2ff}$, deducing likely transport errors in an example flight. Although ACT-America $CO_{2ff}$ signals were attenuated due to broad source regions sampled, results illustrate the value of $\Delta^{14}CO_2$ sampling and observation-based methodologies for regional $CO_2$ flux attribution and evaluation and improvement of modeled $CO_2$.

## 1 Introduction

Anthropogenic carbon dioxide ($CO_2$) emissions have driven more than a 50% increase in atmospheric $CO_2$ abundances globally since pre-industrial levels, despite almost half of these emissions being removed from the atmosphere by terrestrial and oceanic reservoirs (i.e., "sinks"; Ballantyne et al., 2012; Friedlingstein et al., 2022). Because total $CO_2$ fluxes encompass biogenic,

oceanic and anthropogenic processes, and regional spatial scales are ones at which carbon mitigation strategies are generally

developed and implemented, it is important that these $CO_2$ component processes be accurately quantified. Fossil fuel emissions represent the bulk of the net atmospheric $CO_2$ flux annually on regional scales for North America (King et al., 2007; USGCRP, 2018). However, while national-scale fossil fuel $CO_2$ fluxes for many countries are likely known to within 10% (Gregg et al., 2009; Andres et al., 2012), fluxes and their variability are less certain on smaller spatial and temporal scales (USGCRP, 2018; Gurney et al., 2021). Even so, fossil fuel fluxes are currently reported with uncertainties up to five times lower than biospheric

$CO_2$ fluxes on regional scales (Hayes et al., 2012; King et al., 2015; USGCRP, 2018).

Atmospheric $CO_2$ and its long-term global growth rate can be directly determined from in situ observations because observed $CO_2$ over large land areas is a function of both fossil fuel emissions and net ecosystem exchange (NEE) with the terrestrial biosphere. Distinguishing fossil fuel from biogenic $CO_2$ fluxes cannot be accomplished from $CO_2$ observations alone (Shiga

et al, 2014; Basu et al, 2016). The radiocarbon to total carbon ratio in atmospheric $CO_2$ ($^{14}$C:C, expressed as $\Delta^{14}CO_2$) has been demonstrated to be a robust and largely unbiased tracer for accurately constraining recently-emitted fossil fuel $CO_2$ ($CO_{2ff}$) into the atmosphere because fossil fuels do not contain $^{14}$C (Levin et al., 2003; Levin and Karstens, 2007; Graven et al., 2009; Vogel et al., 2010; Miller et al., 2012; Turnbull et al., 2006, 2011a, 2011b, 2015).  By isolating this fossil fuel component and its variability relative to background levels, the remaining variability in $CO_2$ can be attributed to biogenic processes.


Inverse models using only $CO_2$ measurements can theoretically be used to disaggregate anthropogenic and biogenic $CO_2$ fluxes when these emissions can be adequately separated in time and space, but given the low density of most regional measurement networks, fossil and biogenic fluxes are co-located, so this disaggregation cannot be achieved in practice (Shiga et al., 2014; Basu et al., 2016). In many situations from regional to global scales, prior fossil $CO_2$ emissions are traditionally pre-specified

and not optimized in $CO_2$ measurement-based inversions (Gurney et al., 2003; Ciais et al., 2010; Schuh et al., 2010; Lauvaux et al., 2011) on the basis that fossil fluxes are much better known than terrestrial biospheric fluxes. Much work has been done to characterize model uncertainty and improve estimates of $CO_2$. At the continental scale and for annual time frames, Feng et al., (2019a) used a forward calibrated model ensemble (Feng et al., 2019b) to show that the uncertainty in simulated North American atmospheric $CO_2$ mole fractions comes primarily from two approximately equivalent sources: fossil fuel emissions

and biogenic $CO_2$ fluxes. At these time scales, these two sources greatly outweighed the uncertainty contributions from atmospheric transport and from continental boundary conditions, and Feng et al. (2019a) note that accurate accounting of fossil fuel emissions uncertainties in models can further improve regional biospheric flux estimates. Brophy et al. (2019) emphasized the need for varying prior fossil fuel emissions estimates in time, alongside needed improvements in the representation of atmospheric transport to accurately represent $CO_2$ fluxes at ~sub-annual and regional scales. Modelers have also relaxed the

assumption of perfectly known fossil $CO_2$ emissions – and have lowered fossil fuel emissions uncertainties – by assimilating $\Delta^{14}CO_2$ measurements (or $CO_{2ff}$) alongside $CO_2$ and carefully controlling systematic model errors and errors in prior flux estimates (Basu et al., 2016, 2020; Fischer et al., 2017).

Using aircraft $\Delta^{14}CO_2$ observations, an effort has been made to evaluate modeled $CO_{2ff}$ and to estimate or verify regional emissions inventories (e.g. Graven et al., 2009; Basu et al., 2020). At the urban scale, studies have also successfully used $\Delta^{14}CO_2$ measurements alongside in situ carbon monoxide (CO) observations from surface- and aircraft-based platforms, to calculate high-resolution proxy estimates of recently-added fossil fuel $CO_2$ to the atmosphere, given well-characterized fossil fuel $CO_2$:CO ratios (Vogel et al., 2010; Turnbull et al., 2011a, 2019; Lauvaux et al., 2020; Wu et al., 2022). As biogenic $CO_2$ emissions and sinks – even within cities – are non-negligible, and ignoring these signals could potentially bias fossil fuel $CO_2$ emission inventories, $\Delta^{14}CO_2$ measurements play an important role in determining biogenically-driven ($CO_2$ photosynthesis and respiration) emissions or sinks (Levin et al., 2003; Lopez et al., 2013; Turnbull et al., 2015; Miller et al., 2020). One tracer that could aid in the understanding of $CO_2$ uptake processes alongside $\Delta^{14}CO_2$ is carbonyl sulfide (OCS), which is taken up by plants but not respired (Montzka et al., 2007; Campbell et al., 2008).

NASA's multi-year Atmospheric Carbon and Transport (ACT)-America Earth Venture Suborbital mission was directed toward improving the accuracy of regional greenhouse gas flux estimates, specifically through refining our understanding and implementation of $CO_2$ and methane ($CH_4$) transport, fluxes and flux uncertainties and background levels in inverse models (Davis et al., 2021). During this mission, five seasonal research flight campaigns were conducted between 2016 and 2019 for evaluation and subsequent improvement of terrestrial carbon cycle models and filling gaps in the eastern U.S. carbon monitoring network. With a strong focus on transport of carbon via weather systems, atmospheric trace gases and meteorological variables were also observed across multiple synoptic cycles, across cold and warm air sectors and from the boundary layer to the upper troposphere. While the ACT mission was not targeting large carbon emissions signals in urban areas as in the abovementioned studies, we focus on carbon flux attribution and model evaluation here. We analyze a spatially dense set of airborne whole-air flask $\Delta^{14}CO_2$ measurements collected during ACT in the Mid-Atlantic U.S. alongside those of other gas-phase species such OCS, a tracer for photosynthetic uptake of $CO_2$, to first distinguish between biogenically-driven and fossil fuel atmospheric $CO_2$ variability relative to background levels. Second, we apply the methodology of previous authors (Levin and Karstens, 2007; Vogel et al., 2010; Turnbull et al., 2011a; Maier et al., 2024a) to the broad, Mid-Atlantic U.S. region, calculating high-frequency fossil fuel $CO_2$ enhancements using in situ carbon monoxide (CO) observations, and investigate the utility of this technique for evaluating modeled atmospheric fossil fuel-$CO_2$ enhancements above background levels for an example ACT research flight along the northeastern corridor.

## 2 Data and Methods

### 2.1 ACT-America flight campaigns

During the ACT mission, the eastern half of the coterminous U.S. was broadly surveyed by two research aircraft: the NASA Langley Beechcraft B-200 King Air and the NASA Wallops C-130 Hercules. Of the Mid-Atlantic, Midwestern and Southern U.S.-focused measurement regions, air samples for $\Delta^{14}CO_2$ measurement were collected only in the Mid-Atlantic region due to proximity to the northeastern U.S. urban corridor and likelihood for higher $\Delta^{14}CO_2$ signals. Sampling in the Mid-Atlantic region focused heavily on the extensive forests and croplands with flights capturing distant urban influence, and a selected few flights specifically targeted the urban northeastern corridor. The first ACT campaign occurred during summer 2016 (July 18th -August 28th) and is omitted from this analysis due to the absence of $\Delta^{14}CO_2$ sampling. Four subsequent ACT campaigns described here occurred during winter 2017 (March 1st-10th), fall 2017 (October 1st-15th), spring 2018 (May 4th-20th), and summer 2019 (July 7th-27th) for a total of 26 flights with atmospheric sampling for $\Delta^{14}CO_2$. Research flights were primarily conducted between 11:00 and 18:00 local time to sample a well-developed atmospheric boundary layer (ABL), through long, level-leg flight transects. Airborne atmospheric sampling during individual research flights was focused at altitudes within the ABL (flight altitudes of 330 m above ground level in most cases) where greenhouse gas abundances are strongly influenced by surface fluxes with occasional sampling in the free troposphere (FT, ~2400 to 9000 m above mean sea level (ASL)) to determine chemistry and isotopic composition of background air against which observed ABL enhancements or depletions are assumed to occur. Because a scientific goal of ACT was to improve the simulation of atmospheric transport of greenhouse gases in regional-scale inversions under real-world conditions, research flights were organized to sample across synoptic weather patterns (Pal et al. 2020). Wei et al., (2021) describe the ACT observational and numerical data products.

### 2.2 Whole-air sample collection and flask radiocarbon measurements

NOAA Programmable Flask Packages (PFPs) were installed on each aircraft for the collection of whole-air samples. Each PFP contains 12, 0.7 L borosilicate glass flasks and for ACT campaigns between 2017 and 2019, these automated systems were connected to a Peltier gas chiller to efficiently dry air samples to less than 1% atmospheric water vapor prior to air sample collection. Flask filling times varied between two (ABL) to four minutes (FT) depending on the altitude of the aircraft. A Programmable Compressor Package (PCP) pressurized each flask to 275 kPa yielding ~ 2 standard liters (L) of air collected per flask. Technical details of these flask sampling systems used during ACT are described in Baier et al. (2020).

For each flight day, roughly six to eighteen total flasks were filled on each aircraft platform between the altitudes of 300-9000 m ASL. The ratio of well-mixed ABL to FT flasks sampled during ACT was generally 5:1. Because approximately 2 L of sample air is required for high precision $\Delta^{14}CO_2$, in addition to the long-lived greenhouse gases ($CO_2$, $CH_4$) and other low concentration gases such as OCS and CO, two PFP flasks were filled in parallel (i.e. 'paired sampling') when sampling for

$\Delta^{14}CO_2$ measurements. Post-flight, PFPs were transferred to the NOAA Global Monitoring Laboratory and the University of Colorado/INSTAAR Stable Isotope Laboratory for measurement. The first flask of each paired sample is analyzed for greenhouse gases, OCS, and a suite of hydrocarbons, halocarbons, and stable isotope ratios ($^{13}C$:$^{12}C$) of $CO_2$ and methane using methods described in Baier et al. (2020), online (https://gml.noaa.gov/ccgg/aircraft/analysis.html) and in Vaughn et al. (2004). Remaining sample air in the first flask after these measurements is combined with the entire second, paired flask for a single $\Delta^{14}CO_2$ measurement. The University of Colorado INSTAAR Laboratory for AMS Radiocarbon Preparation and Research (NSRL) conducts the $CO_2$ extraction from flask sample air. Pure $CO_2$ is archived until near the time of $\Delta^{14}CO_2$ measurement, at which time the pure $CO_2$ is graphitized to pure carbon "targets" and these graphitized samples are transferred to the University of California at Irvine for high-count accelerator mass spectrometry (AMS) $^{14}C$ measurement (Turnbull et al., 2007; Lehman et al., 2013). In total, 380 $^{14}CO_2$ samples were analyzed throughout the ACT campaigns: 87 in winter 2017, 100 in fall 2017, 104 in spring 2018 and 89 in summer 2019.

## 2.3 Radiocarbon-based $CO_2$ partitioning

Radiocarbon ($^{14}C$) is produced naturally in the troposphere and stratosphere via cosmic ray-induced reactions between neutrons and atmospheric nitrogen ($N_2$) (Libby, 1946). Measurements are presented as $\Delta^{14}CO_2$ in units of per mil (‰), or the part per thousand deviation of the measured $^{14}C$:C ratio from that of the international measurement standard material, after correction for mass dependent fractionation and radioactive decay since the date of collection (Stuiver and Polach, 1977). Given that the $^{14}C$ half-life is approximately 5730 years (Godwin, 1962), and that fossil fuel carbon has been isolated from the atmosphere for millions of years, the radiocarbon content of fossil fuels is zero and therefore $\Delta^{14}CO_{2ff}$ is -1000 ‰ on the delta ($\Delta$) scale. Therefore, addition of fossil fuel $CO_2$ to the atmosphere produces a quantitative reduction in atmospheric $\Delta^{14}CO_2$ over and downwind of emissions sources. Paired observations of $\Delta^{14}CO_2$ and $CO_2$ in flasks sampled in the Mid-Atlantic region can thereby provide a robust method for distinguishing between terrestrial biogenic ($CO_{2bio}$) and fossil fuel ($CO_{2ff}$) $CO_2$ sources given a known background. The $CO_2$ mass balance over land is:

$$CO_{2obs} = CO_{2bg} + CO_{2ff} + CO_{2bio} \qquad (1)$$

Expanding $CO_{2bio}$, which results from Net Ecosystem Exchange, into a sum of photosynthetic (*photo*) and respiration (*resp*) fluxes ($CO_{2resp} + CO_{2photo}$) and adding the respective isotopic signatures ('$\Delta$', for $\Delta^{14}CO_2$) for all terms gives Equation (2):

$$\Delta_{obs}CO_{2obs} = \Delta_{bg}CO_{2bg} + \Delta_{ff}CO_{2ff} + \Delta_{resp}CO_{2resp} + \Delta_{photo}CO_{2photo} + N^{14}C_{nuc}/R_{std}, \qquad (2)$$

where the individual budget terms are the product of the isotopic ratio and $CO_2$ mole fraction, which is a conserved quantity (Tans et al., 1993). Here, *obs* is the observed atmospheric mole fraction of $CO_2$ in the ABL and is a function of variations in

fossil fuel (*ff*) and biogenic (*bio*) $CO_2$ differences from a varying $CO_2$ background (*bg*) level. We note that, an oceanic term is sometimes written in Equation (1), but we assume that the dominant oceanic $CO_2$ influence is encompassed in the background (*bg*) term. Photosynthesis and respiration contributions are stated separately in the isotopic mass balance (Eq. 2), because they carry different isotopic signatures. This isotopic "disequilibrium" between photosynthetic and respiration fluxes arises from the rapidly decreasing ($\sim$ -5 ‰ $yr^{-1}$) background $\Delta^{14}CO_2$ (e.g., Lehman, 2013; Graven et al., 2020). $\Delta_{resp}$ is more positive than $\Delta_{photo}$ because $\Delta_{resp}$ is associated with biospheric $CO_2$ fixed by photosynthesis when the ambient atmospheric $\Delta^{14}CO_2$ (i.e. $\Delta_{bg}$) was higher. $\Delta_{photo}$ is assumed equal to $\Delta_{bg}$ because the '$\Delta$' notation accounts for mass-dependent fractionation processes such as those occurring during photosynthesis. Turnbull et al. (2009) note that this assumption is valid in the limit that time and space between background and observation tends to zero. There is also a small impact on $\Delta_{obs}$ from pure $^{14}C$ emitted from certain nuclear reactors, which is added as the last term in Equation 2 (Graven and Gruber, 2011). Because this term is significantly smaller than all other terms in Equation (1), it is omitted there. Here $N$ is the mass dependent $\Delta$ normalization factor, which is close to 1 (Basu et al., 2016) , $^{14}C_{nuc}$ is the mole fraction of $\Delta^{14}CO_2$ resulting from the $^{14}C$ flux from reactors and $R_{std}$ is the standard material $^{14}C/C$ ratio.

Combining Equations (1) and (2) results in:

$$CO_{2ff} = \frac{CO_{2obs}(\Delta_{obs}-\Delta_{bg})}{(\Delta_{ff}-\Delta_{bg})} - \frac{CO_{2resp}(\Delta_{resp}-\Delta_{bg})}{(\Delta_{ff}-\Delta_{bg})} - N^{14}C_{nuc}/R_{std}/(\Delta_{ff}-\Delta_{bg}). \tag{3}$$

The variables in Eq. (3) are either known or can be measured or calculated directly. Historically, $CO_{2ff}$ has been approximated by only the first term on the right-hand side of Equation (3) (Levin et al., 2003). Disequilibrium and nuclear fluxes both add $^{14}C$ to the atmosphere. A correction ($CO_{2corr}$) is made to $CO_{2ff}$ (Turnbull et al., 2009) to unmask these effects and reveal the full magnitude of the fossil fuel-$CO_2$ emissions signal, making it somewhat convenient to rewrite Equation (3) as:

$$CO_{2ff} = \frac{CO_{2obs}(\Delta_{obs}-\Delta_{bg})}{(\Delta_{ff}-\Delta_{bg})} - CO_{2corr} . \tag{4}$$

Flask ABL measurements for $CO_{2obs}$ are defined as those samples collected at altitudes less than 1500 m ASL for all ACT flight campaigns. ABL samples typically contain the strongest regional flux signatures, and thus are likely to yield the largest $\Delta^{14}C$ signals. Conversely, we determine daily $\Delta_{bg}$ and $CO_{2bg}$ values from measurements of $^{14}CO_2$ and $CO_2$ in flasks sampled at altitudes greater than 4000 m ASL for all flight campaigns to capture air that is within the well-mixed FT in all seasons. The FT data, while perhaps not always representative of the true, local-scale background signature (e.g. Turnbull et al., 2015), is expected to provide a reliable estimate of the regional- to continental-scale background (Baier et al., 2020). Here, $\Delta_{bg}$ in Eq. 2 is derived as the daily mean FT $\Delta^{14}CO_2$, typically between 4000 and 9000 m ASL. All flask FT measurements are filtered to remove samples with high CO (above three standard deviations from a smooth curve fit), which can indicate the potential

influence of local pollution on continental background values. Similarly, *CO2bg* and other trace gas background mole fractions were derived as mean mole fractions of FT values for these species. For the ACT study, $CO_{2corr}$ is derived for each flask sample by convolving estimated gridded monthly disequilibrium fluxes of $\Delta^{14}CO_2$ from the terrestrial biosphere and fluxes from nuclear reactors with surface influence functions for each flask sample location (i.e., for each flask "receptor", per below).

195

Surface influence functions (referred to as "footprints"), representing the sensitivity of air parcels at a particular receptor location to upwind surface fluxes (in units of ppm $m^2\,s\,\mu mol^{-1}$), are derived by first using the Hybrid Single-Particle Lagrangian Integrated Trajectory (HYSPLIT) model (Draxler and Hess, 1997) running 500 randomly perturbed, 10-day back trajectories for all ABL flask receptors. Back trajectories are driven by 27-km horizontal resolution Weather Research and Forecasting (WRF) meteorological fields, nudged to the ERA-5 reanalysis meteorology within the ACT-America North American model domain (Feng at al., 2021b, see Section 2.4). Individual footprints are calculated for all 500-particle trajectory ensemble members based on their residence time over a given gridded area (Lin et al, 2003) within the surface boundary, defined as a vertical column of air in each $1^{\circ}$ x $1^{\circ}$ grid cell between 125 m AGL and 50% of the WRF-calculated atmospheric boundary layer height (Fig. 1). Referring to Eqns. (1) and (2), though there exists a footprint signature on flask samples during fall 2017 and spring 2018, this influence is relatively small. Disequilibrium fluxes from the terrestrial biosphere are derived by convolving impulse-response functions from the CASA biogeochemical model (Thompson and Randerson, 1999) with the atmospheric history of $^{14}C$ (Miller et al. 2012; LaFranchi et al., 2016, with updates in Zhou et al., 2020) while nuclear reactor fluxes are obtained from Graven and Gruber (2011), effectively assuming regional nuclear $^{14}CO_2$ has not changed significantly over time through 2019. These gridded fluxes (monthly, time-dependent in the case of disequilibrium flux) are converted to the two components of $CO_{2corr}$ via convolution with footprints derived from HYSPLIT and are calculated individually for each ABL flask sample pair for which $CO_{2ff}$ is derived using Equation (4). We note that the calculated magnitude of $CO_{2corr}$ attributable to nuclear reactor emissions in the Mid-Atlantic U.S. is $0.25 \pm 0.44$ ppm $CO_2$ and does not exhibit a seasonal pattern as it is assumed constant annually. Graven and Gruber (2011) have been used at the time that this publication was written, but updates to nuclear reactor fluxes have been published in Zazzeri et al. (2023). The magnitude of $CO_{2corr}$ attributable to the terrestrial biosphere in the Mid-Atlantic U.S. is approximately $0.10 \pm 0.04$ ppm $CO_{2ff}$ ($1\sigma$) during the winter deployment and exhibits a seasonal cycle with a maximum during the summer of $0.70 \pm 0.28$ ppm $CO_{2ff}$ ($1\sigma$), indicating similar seasonal behavior as that calculated by Miller et al. (2012) for the Mid-Atlantic U.S. In total, the average magnitude of $CO_{2corr}$ ($0.8 \pm 0.6$ ppm) calculated here to correct $CO_{2ff}$ in this work is roughly comparable that first described in Turnbull et al. (2006).

### 2.4 Forward model $CO_{2ff}$ using the PSUWRF model

A separate, Eulerian 27 km horizontal resolution implementation of WRF-Chem version 3.6.1 run (Feng et al., 2021a,b) was implemented by the Pennsylvania State University (hereby called PSUWRF) with 50 vertical levels from the surface to 50 hPa, with 29 of these vertical levels in the lowermost 2 km. PSUWRF was developed and implemented during the ACT time

period (2016-2019) to simulate atmospheric transport of $CO_2$ component fluxes along all flight tracks for each seasonal campaign. Note that, while the Lagrangian HYSPLIT model mentioned above to calculate $CO_{2corr}$ in Equation 4 for each flask measurement also used WRF for input meteorological fields, this was not identical to PSUWRF used for the Eulerian simulations. PSUWRF model runs were implemented prior to this manuscript development and could not be re-run, but both WRF versions used similar setups and thus we expect there to be no major inconsistencies between the use of these two different versions for calculating $CO_{2ff}$ using flask or model output.

PSUWRF separately simulates the contributions of $CO_2$ boundary conditions, and biogenic, ocean, fossil fuel and biomass burning $CO_2$ fluxes to total $CO_2$ at any time and location in the model domain (Feng et al., 2019a,b). The microphysics, PBL and cumulus parameterization schemes used in this run are Thompson microphysics, the Mellor-Yamada-Nakanishi-Nino version 2 (MYNN2), and the Kain-Fritsch schemes, respectively (Feng et al., 2019a). As the ACT campaigns spanned multiple years, $CO_2$ oceanic, fossil fuel and biomass burning fluxes, and total $CO_2$ boundary conditions come from CarbonTracker version 2017 (CT2017) through 2017 and CT-NRTv2019-2 (Jacobson et al., 2020) afterwards. CT2017 and CT-NRTv2019-2 together are called 'CT' for simplicity. The CT fossil fuel tracer typically is derived from an average of the ODIAC and Miller emissions datasets. As both datasets have very similar global and national fossil fuel emissions totals, but include differences spatial and temporal emissions distributions, an average of the two is used in CT to optimize the mapping of these emissions (Jacobson et al., 2020). In PSUWRF, ODIAC and Miller datasets were run separately for initial model runs between 2016-2018 to experimentally investigate potential differences between the two; however, only small differences within ~2 ppm $CO_{2ff}$ are seen between the two and are not expected to create inconsistencies in $CO_{2ff}$ simulated by PSUWRF between 2016-2019. As such, ODIAC and Miller are averaged to create a single flux product and to simplify model runs as in CT for 2019. In PSUWRF, simulated fossil fuel $CO_2$ ($CO_{2ffmod}$) is calculated using the same approach as flask $CO_{2ff}$, where average FT values of this tracer are subtracted from ABL values. The PSUWRF model transport is nudged to ERA5 reanalysis data to improve the depiction of atmospheric transport relative to ACT in-flight meteorological observations (Gerken et al., 2021). PSUWRF model output was extracted to ACT research flight track locations at hourly resolution for comparison and evaluation relative to high-frequency, measurement-based $CO_{2ff}$ described in Section 2.5.

**2.5 Calculation of "pseudo" $CO_{2ff}$**

As in previous studies (Levin and Karstens, 2007; Vogel et al., 2010; Turnbull et al., 2011a, Maier et al., 2024a), we employ in situ CO measurements for estimating $CO_{2ff}$ (referred to as "pseudo-$CO_{2ff}$" and denoted as $CO_{2ff}'$) at high temporal resolution. ACT in situ CO measurements were made by cavity ring-down spectroscopy at 0.4 Hz, drift-corrected during flight and calibrated once weekly using gas standards on the World Meteorological Organization (WMO) X2014A CO scale, just as NOAA flask-based measurements (Wei et al., 2020; DiGangi et al., 2021). After calibration, CO data were averaged to 0.2 Hz, the maximum frequency of $CO_{2ff}'$. We calculate $CO_{2ff}'$ using Equation (5) by applying a median ratio of CO to $CO_{2ff}$ ($R_{CO}$)

calculated strictly from flask samples collected each day (typically from 5-12 flask samples), and apply this value to the in situ
CO time series collected on that day's ACT research flight:

$$CO_{2ff}' = \frac{CO_{obs}' - \overline{CO'_{bg}}}{R_{CO}}.$$    (5)

Here, $CO_{obs}'$ are 0.2 Hz ABL (below 1500 m ASL) in situ CO dry mole fraction observations and $\overline{CO'_{bg}}$ is the mean daily CO
mole fraction observed in the FT (above 4000 m ASL) for each flight day. $R_{CO}$ is calculated as median value of all flask
measurements each day, $i$:

$$R_{CO} = \frac{[CO_{obs,flask} - \overline{CO_{bg,flask}}]_i}{CO_{2ff,flask\_i}}.$$    (6)


A median $R_{CO}$ is used each day due to difficulty with respect to calculating seasonal regression slopes with low correlations
(seasonal $R^2$ between 0.08-0.33) in the flask CO enhancement and $CO_{2ff}$ data. In calculating daily median $R_{CO}$, it may be
possible to capture some spatial variability of research flight observations rather than ignoring this variability and rely on
average seasonal values. However, we acknowledge in Section 3.3.1 that this methodology, given low signal to noise ($\Delta^{14}CO_2$
precision) in the ACT $CO_{2ff}$ data, could create anomalous variability in $R_{CO}$ and is one of the largest sources of uncertainty in
this $CO_{2ff}'$ calculation (as first presented in detail in Maier et al., 2024b) and a limitation to this analysis.. In situ FT
measurements are filtered to remove high CO or local emissions influence. Similar to above, data with CO greater than three
standard deviations from a smooth curve fit to background values are filtered. Each of these terms is derived exactly as the
analysis of flask $CO_{2ff}$ above. Due to the limited number of flasks sampled on each flight, $R_{CO}$ is calculated as a daily median
ratio of CO enhancements to $CO_{2ff}$ from flask samples, and we assume that this ratio is representative of the entire flight region
each day. In the analysis that follows, we assess the utility of this "pseudo-$CO_{2ff}$" method for determining fossil fuel $CO_2$ in
evaluating modeled $CO_{2ff}$ over larger regions where routine, high-density discrete flask sampling is not available. In this work,
$CO_{2ff}'$ is calculated only for flights when flask measurements were sampled and analyzed for $\Delta^{14}CO_2$ in order to reduce errors
associated with assumptions in $R_{CO}$ spatial and temporal variability.

**2.6 $CO_{2ff}$ uncertainty derivation**

The total estimated uncertainty in $CO_{2ff}$ is determined by a propagation of estimated uncertainties for individual budget terms
through Equation (4). Flask-based $CO_{2obs}$ and $\Delta_{obs}$ measurement uncertainties are 0.1 ppm and 1.8‰, respectively. Here $\Delta_{ff}$ is,
by definition -1000‰, and carries no uncertainty. The uncertainty in $\Delta_{bg}$ is calculated through propagation of error in a) $\Delta_{obs}$
and b) the mean diurnal standard deviation of FT $\Delta^{14}CO_2$ observations ($\Delta_{bg}$) for each ACT flight. The relative uncertainty in
$CO_{2corr}$ associated with biospheric disequilibrium and nuclear reactor fluxes is assumed to be 50% for each term.  Daily

aggregated $CO_{2ff}$ sample uncertainties are determined by using a 1,000-member Monte Carlo simulation that propagates these uncertainties through Equation (4) to provide an estimate of the flask-derived $CO_{2ff}$ uncertainty per research flight. The average campaign-wide $CO_{2ff}$ uncertainty (from the inner 68% confidence interval of the 1000-member $CO_{2ff}$ normal distribution per research flight) is 1.22 ppm.


Similar to above, we calculate an average campaign uncertainty in $CO_{2ff}'$ of 4.96 ppm (from the inner 68% confidence interval in $CO_{2ff}'$ of a 1,000-member Monte Carlo approach for each flight day) using Equation (5). We estimate daily uncertainties by conservatively incorporating the CO dry mole fraction measurement precision on 0.2 Hz measurements (5 ppb), the uncertainty in daily background CO from $CO_{obs}$ and the daily standard deviation in FT CO, and the uncertainty in $R_{CO}$ from the width of

a normally-distributed 68% confidence interval about average $R_{CO}$ values.

The uncertainty in both $CO_{2ff}$ and $CO_{2ff}'$ is higher for ACT research flights due to the uncertainty caused by using a single CO or $\Delta^{14}CO_2$ background value for each flight, when in truth, different sectors of a research flight (e.g. the cold and warm sectors during frontal crossing flights) could experience different background conditions or could be influenced by deep convective

mixing, violating the assumption that the continental background value is described by the FT value. Elevating the altitude threshold at which background values are defined (i.e. 4000 m ASL) can better isolate continental background air from lower tropospheric air that has been mixed with local emissions sources. However, using a higher threshold altitude to define background values can also lead to higher uncertainties in air that is considered as a background for local ABL measurements due to the fact that air at or above 4000 m ASL may have experienced longer-range transport.

**3 Results and Discussion**

**3.1 Mid-Atlantic $\Delta^{14}CO_2$ background in general agreement with NOAA aircraft network**

Figure 2a shows the vertical distribution of $\Delta^{14}CO_2$ measured for all seasonal campaigns between winter 2017 and summer 2019, highlighting this difference and indicating generally lower values and significantly greater variation relative to the FT throughout the bottommost troposphere due to surface emissions of $^{14}$C-free fossil fuels.


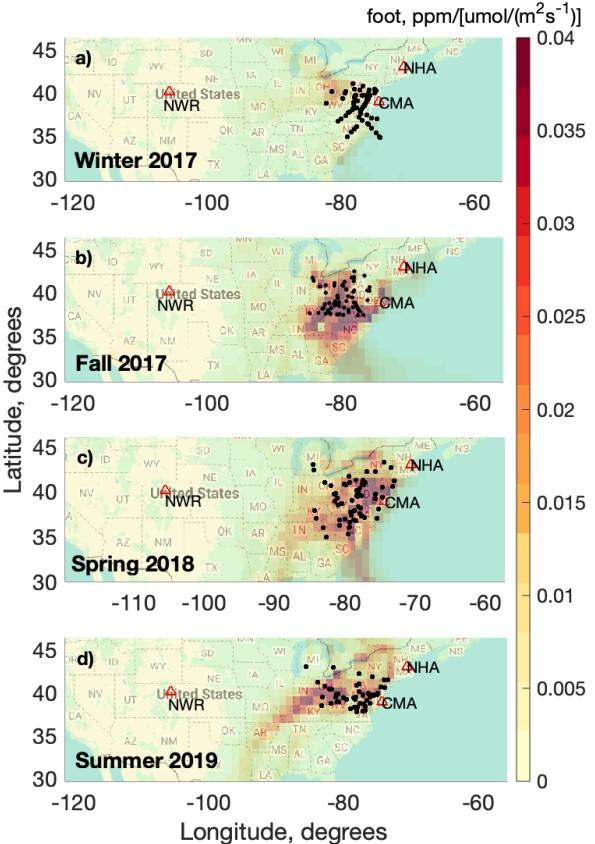

**Figure 1. Surface influence for ABL $\Delta^{14}CO_2$ samples measured for ACT seasonal deployments in a) winter 2017, b) fall 2017, c) spring 2018, and d) summer 2019. ABL flask sample locations are denoted in black, with GGGRN Aircraft Program network sites indicated at Niwot Ridge, CO (NWR), Cape May, NJ (CMA), and Portsmouth, NH (NHA). Plot maps are produced using Google Maps API (©Google Maps).**

Figure 2b shows the decrease in atmospheric $\Delta^{14}CO_2$ in the FT over time during ACT and within the broader NOAA Global Greenhouse Gas Reference Network (GGGRN) due to global-scale addition of fossil fuel $CO_2$ and justifies the use of ACT FT for a definition of background atmospheric conditions. As expected, values of $\Delta_{bg}$ throughout the upper FT are relatively homogenous during each campaign and are in rough agreement with other FT flask $\Delta^{14}CO_2$ measurements within the GGGRN
(Sweeney et al., 2015; Miller et al., 2012) such as Cape May, NJ (CMA) and Portsmouth, NH (NHA). For fall and spring seasons, ACT FT $\Delta^{14}CO_2$ is slightly higher than that in the GGGRN sites, including the upwind, high-altitude (3000-4000 MASL) Niwot Ridge, CO (NWR) site. During ACT, FT samples are frequently obtained from higher altitudes than the GGGRN $\Delta^{14}CO_2$ paired flask samples with fixed collection altitudes, resulting in the potential for ACT FT air to originate from different latitudes or altitudes where influence from cosmogenically-produced $^{14}CO_2$ could explain deviations of ~ 2‰
above GGGRN $\Delta^{14}CO_2$ observations (Turnbull et al., 2009). Early in the ACT winter 2017 campaign, FT $\Delta^{14}CO_2$ is lower than the mean values in the GGGRN (Fig. 2b), and is accompanied by higher than average CO mole fractions. This result potentially indicates a small influence from local pollution sources that are depleted in FT $\Delta^{14}CO_2$. Nonetheless, these ACT FT data agree with (± 1σ standard deviation) GGGRN continental background FT values and are included in this analysis.

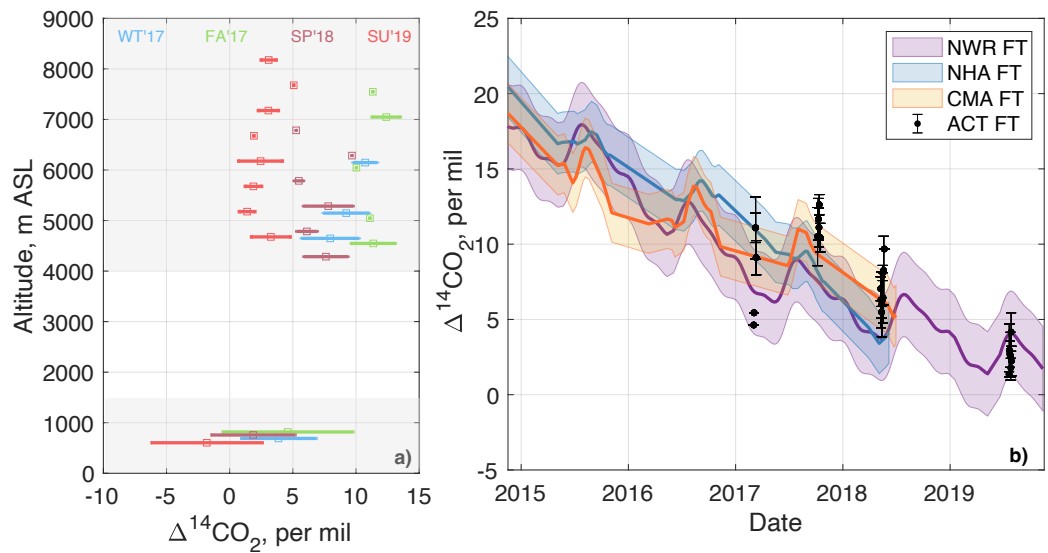

**Figure 2. a): Vertical distribution of $\Delta^{14}CO_2$ during all ACT campaigns in the Mid-Atlantic U.S. ABL values shown are the average of all observations below 1500 m ASL (values are separated vertically for clarity) for each seasonal campaign while horizontal bars represent their 1σ standard deviation in time. FT values (altitudes greater than 4000 m ASL) shown are 500-m binned averages and their 1σ standard deviation in $\Delta^{14}CO_2$ for each 500 m altitude bin. Shaded regions represent ABL and FT definitions. b): Daily mean ACT FT $\Delta^{14}CO_2$ background measurements (± 1σ standard deviation shading) compared to NOAA GGGRN aircraft network $\Delta^{14}CO_2$ FT (~4000 m ASL) measurements. GGGRN data are seasonal (3-month averaged) aircraft FT $^{14}CO_2$ obtained offshore of Cape May, NJ (CMA) and offshore of Portsmouth, New Hampshire (NHA) and surface $\Delta^{14}CO_2$ for the high-altitude site, Niwot Ridge, CO (NWR) at ~3500 m ASL.**

Figure 3a highlights the spatial extent of flask sampling during the ACT mission and density of flask sampling over the four seasonal campaigns with an average $\Delta^{14}CO_2$ ABL-FT difference of -5‰. Using Equation (4), $CO_{2ff}$ is calculated for all flask samples, and general agreement between largely negative $\Delta^{14}CO_2$ ABL signals and higher $CO_{2ff}$ signals are seen throughout ACT (Fig. 3b). ACT observed $CO_{2ff}$ ranges from -0.8 to 15.5 ppm, noting that negative $CO_{2ff}$ values are not physically realistic but can occur due to the measurement uncertainty of $\Delta^{14}CO_2$, by using an inappropriate $\Delta^{14}CO_2$ background value, or by underestimating the $CO_{2corr}$ term in Eq. 4. Compared to other surface and aircraft observations of $CO_{2ff}$ throughout the U.S., the range of ACT $CO_{2ff}$ is comparable to $CO_{2ff}$ measured in other studies in the Mid-Atlantic U.S. (Miller et al., 2012: -3-13 ppm $CO_{2ff}$), downwind of Sacramento, CA (Turnbull et al., 2011a: 2.4-8.6 ppm $CO_{2ff}$) and in Colorado urban regions (Graven et al., 2009: 0-20 ppm $CO_{2ff}$). Maximum values of $CO_{2ff}$ measured during ACT were however much lower than those over the densely populated urban area of Los Angeles, CA (Miller et al., 2020: -1.3-48.4 ppm $CO_{2ff}$) due to its more rural sampling

focus. Few ACT flight legs were aimed at capturing local, urban-scale $CO_{2ff}$ signals. As such, the analysis below requires

averaging of these signals as a more robust way to qualitatively analyze the ACT flask $CO_{2ff}$ data.

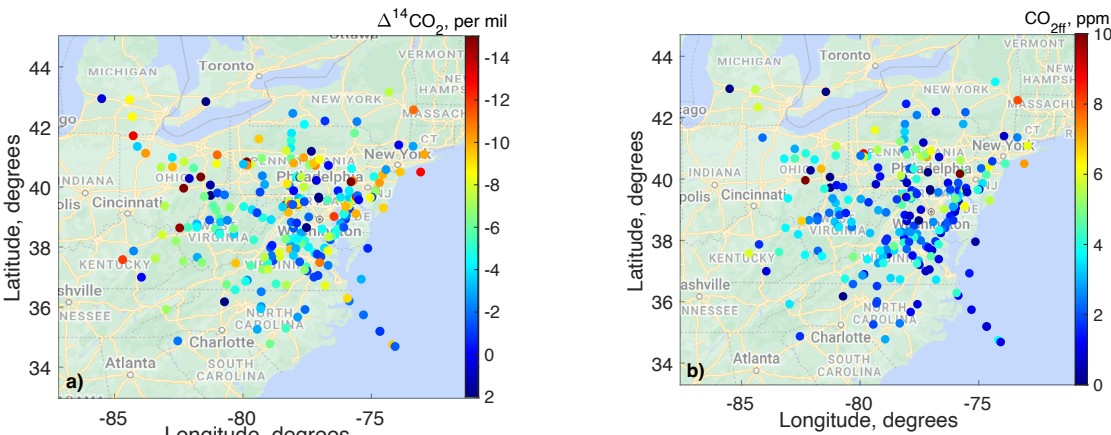

**Figure 3: a) $\Delta^{14}CO_2$ ABL-FT differences for all seasons in the Mid-Atlantic U.S. Note that the color bar is reversed to indicate warmer colors for more negative $\Delta^{14}CO_2$. b) ABL $CO_{2ff}$ calculated for the same flask samples shown in a) for all seasons in the Mid-Atlantic U.S. Plot maps are produced using Google Maps API (©Google Maps).**

### 3.2 Partitioning of seasonal $CO_{2tot}$ indicates biogenic-driven variability

For consistency throughout ACT seasonal campaigns, ABL values for all species in the subsequent analysis are reported as an ABL-FT difference. Figure 4 shows $CO_{2ff}$, alongside biogenic $CO_2$ ($CO_{2bio}$) calculated from Equation (1), displayed by month

of the year for ACT deployments in winter (2017), spring (2018), summer (2019) and fall (2017) in the Mid-Atlantic United States. Total ($CO_{2tot}$), fossil fuel and biogenic $CO_2$ are shown to indicate relative contributions to the observed $CO_{2tot}$ variability (Eq. 1). $CO_{2tot}$ observed in the ABL varies seasonally; while most often positive for flights conducted in late fall and winter, $CO_2$ depletion is seen during summer as expected due to net photosynthetic uptake by the terrestrial biosphere (Tans et al., 1990). Seasonal changes in ABL $CO_{2tot}$ were driven by $CO_{2bio}$, which varied from -19.7 ppm during summer flights to 16.2

ppm in late fall (Fig. 4). In early fall, negative $CO_{2bio}$ was observed (Fig. 4) from strong surface influence and net uptake in southwestern Virginia (Fig. 1). For the remainder of the fall campaign, flights indicated generally positive $CO_{2bio}$ with net respiration and weaker surface influence. $CO_{2ff}$ has a lower contribution to $CO_{2tot}$ variability during ACT in the Mid-Atlantic region from one campaign to the next, with a positive contribution of 3.3 ± 2.0 ppm. Given the small number of flights shown over three years, it should be noted that the seasonality in $CO_{2ff}$ may not represent true climatological $CO_{2ff}$ variability and

should not be interpreted in terms of uniform fossil fuel emissions because there is likely seasonal variation in ABL depth, which can counteract seasonally-varying emissions.

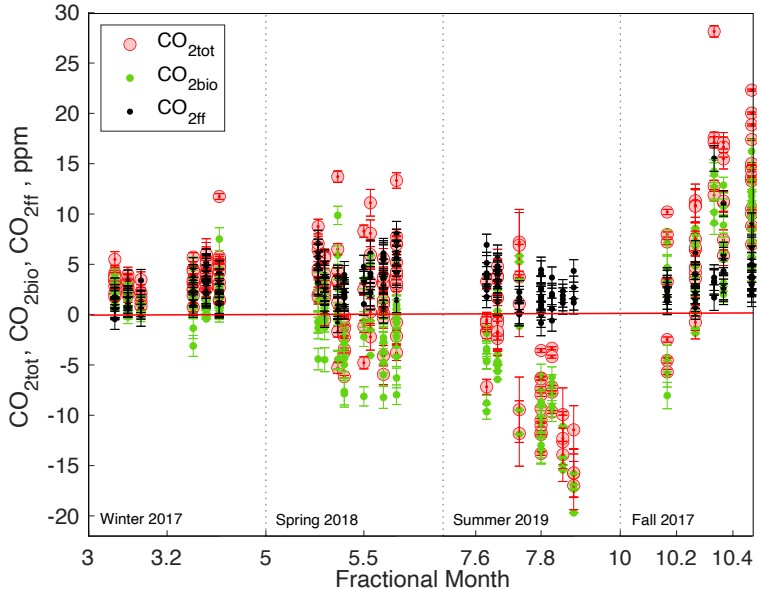

**Figure 4. Calculated total CO₂ (CO₂tot), partitioned into biogenic CO₂ (CO₂bio) and fossil fuel CO₂ (CO₂ff) components using Eq. (4) in winter 2017; spring 2018; summer 2019; and fall 2017. As seasonal campaigns were not conducted consecutively within a single year, we examine variability according to a "climatological" year. Note that the x-axis is intentionally non-linear to show intra-day variability. One sigma error bars are shown for CO₂tot, CO₂bio, and CO₂ff.**

The flask sampling region for this study encompasses rural to suburban regions, reducing signal to noise in $\Delta^{14}CO_2$ and complicating the interpretation of $CO_{2tot}$ observed due to a wide mixture of surface sources (and sinks). As $CO_{2tot}$ is largely controlled by $CO_{2bio}$, we correlate (using Pearson's $R^2$ values) and regress $CO_{2tot}$ with $CO_{2ff}$ and with $CO_{2bio}$ (see Table S1 of supplementary material). Regressions between $CO_{2bio}$ and $CO_{2tot}$ result in slopes between 0.86 and 0.92 for all seasons. Strong correlations between $CO_{2bio}$ and $CO_{2tot}$ exist from spring through fall with $R^2$ values between 0.73 and 0.92. Somewhat strong relationships between $CO_{2tot}$ and $CO_{2bio}$ were found during the March winter 2017 campaign with a regression slope close to 1 but relatively weak correlation ($R^2 \sim 0.3$), which could partially be explained by small $CO_{2ff}$ signals, but also indicate that observed CO₂ had a non-negligible influence from biogenic CO₂ exchange as was also found in previous studies (Potosnak et al., 1999; Turnbull et al., 2006; Miller et al., 2012; Baier et al., 2020). Correlations between $CO_{2tot}$ and $CO_{2ff}$ are weak, with associated regression slopes highest during winter and smaller slopes observed for the spring, summer, and fall deployments when ecosystem CO₂ fluxes are higher.

The fraction of negative $CO_{2tot}$ during biogenically active months in spring (May), summer (July), and fall (October) are 28%, 62%, and 4%, respectively (Table S1). However, when examining the fraction of negative $CO_{2bio}$ (indicating net $CO_2$ uptake), these percentages are much larger at 57%, 82%, and 8%, respectively. The difference between negative $CO_{2tot}$ and negative $CO_{2bio}$ fractions of 20-30% during months when the ecosystems are more active highlights the importance of using $\Delta^{14}CO_2$ to enable partitioning of $CO_{2tot}$ into fossil fuel and biogenic components; this would not be possible when considering $CO_{2tot}$ alone, as a portion of this flux component would be masked by fossil fuel emissions.

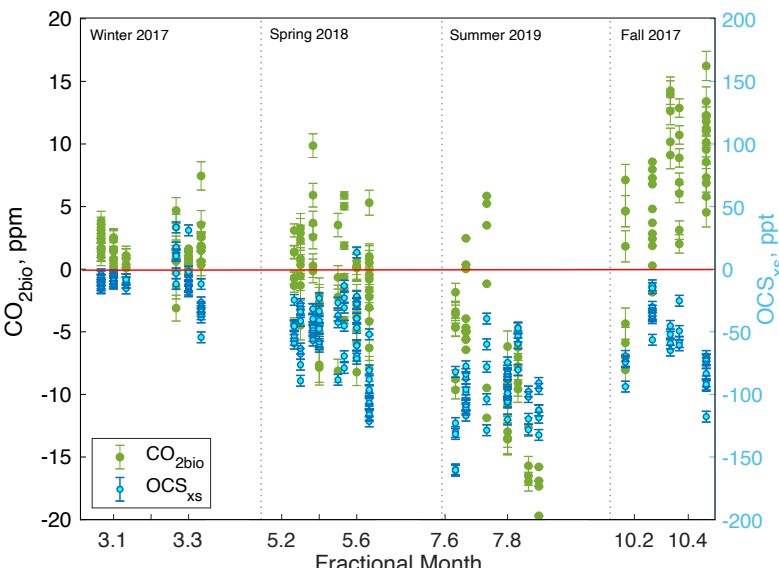

**Figure 5. ABL biogenic $CO_2$ ($CO_{2bio}$) and OCS ABL-FT difference ($OCS_{xs}$) in winter 2017; spring 2018; summer 2019; and fall 2017. As seasonal campaigns were not conducted consecutively within a single year, we examine variability according to a "climatological" year. Note that the x-axis is intentionally non-linear to show intra-day variability. One sigma error bars are shown for $CO_{2bio}$ and $OCS_{xs}$.**

We also examine carbonyl sulfide ABL-FT differences ($OCS_{xs}$) alongside $CO_{2bio}$ as an additional tracer for biogenic processing as OCS is taken up by plants similarly to $CO_2$ but not respired (Montzka et al., 2007; Campbell et al., 2008, Fig. 5). In Figure 5, we observe a strong positive correlation with $CO_{2bio}$ and $OCS_{xs}$ when combining all seasonal ACT data except fall where $CO_{2bio}$ is either negative or weakly positive (see Fig. S1a of supplementary material), consistent with photosynthesis explaining negative $CO_{2bio}$ values. A modest positive correlation (R = 0.54) is seen when also including fall ACT data. Including or excluding the winter data, given the insignificant slope between $OCS_{xs}$ and $CO_{2bio}$ (Table S1), does not affect this correlation.

For fall 2017, the overall correlation between $CO_{2bio}$ and $OCS_{xs}$ is low and not significant (Table S1). However, statistically significant correlations are seen between negative $CO_{2bio}$ and $OCS_{xs}$ in early fall due to continued uptake of $CO_2$ by the terrestrial biosphere (Fig. 5). A moderate negative correlation is observed between $OCS_{xs}$ and $CO_{2bio}$ in late fall after net respiration is seen to occur (Fig. 5, Fig. S1b). During this time, $CO_{2bio}$ of up to ~16 ppm occurs alongside relatively significant

OCS$_{xs}$ of almost -120 ppt. The net negative correlation during fall between CO$_{2bio}$ and OCS$_{xs}$ indicates that, while CO$_2$ uptake could still be occurring, the magnitude of this process is not large enough to offset respiration. Similar to ACT, clear differences in the amplitude and phases of OCS and CO$_2$ cycles were first described in Montzka et al. (2007) and in other, previous studies (Kuai et al., 2022, Ma et al., 2023). Furthermore, many OCS studies have found several instances where ecosystem OCS uptake is decoupled from GPP, including nighttime processes (Commane et al., 2015; Kooijmans et al., 2017; Hu et al., 2021), uptake from soil (Whelan et al., 2022) and/or from senescing or decaying fall vegetation (Sun et al., 2016; Rastogi et al., 2018). Further studies could utilize collocated measurements such as OCS and radiocarbon-based CO$_{2bio}$ to evaluate a) the relationship between OCS and CO$_2$ cycles during the transition from net photosynthesis to net respiration and b) regional model GPP and respiration fluxes. Further, we note that correlations between negative CO$_{2tot}$ and OCS$_{xs}$ weaken in the early fall due to proportionally high fossil fuel emissions, providing insufficient information about GPP. Parazoo et al. (2021) found that models underestimate observed CO$_{2tot}$ during ACT, and have decreased fidelity in reproducing GPP inferred from observations throughout the U.S. Since stronger relationships emerge between OCS$_{xs}$ and CO$_{2bio}$ than OCS$_{xs}$ and CO$_{2tot}$, again, using OCS and radiocarbon-based CO$_{2bio}$ could further inform and constrain these model processes.

### 3.3 Example Case: "Pseudo-CO$_{2ff}$" as a product for evaluating model error

We use CO$_{2ff}$ as a model "transport tracer", to examine model errors as CO$_{2ff}$ fluxes are assumed to be relatively well-known (Turnbull et al., 2009). Well-known emissions and well-measured mole fractions directly tracing those emissions allow us to evaluate the atmospheric tracer transport that connects the two. Here, we compare a "pseudo-CO$_{2ff}$" or high-frequency CO$_{2ff}$' (Equation 5) to CO$_{2ffmod}$ simulated using the PSUWRF forward model, based on Miller fossil fuel emissions (Jacobson et al., 2020). This CO$_{2ff}$' product being high-frequency can also capture more diurnal variability in fossil fuel CO$_2$, and therefore evaluate modeled fossil fuel CO$_2$ more thoroughly, than measurements made using discrete flask samples.

### 3.3.1 Variability and uncertainty of R$_{CO}$ and CO$_{2ff}$

Our CO$_{2ff}$' estimation relies on the assumption that the emission ratio of CO to CO$_{2ff}$ (R$_{CO}$) is constant for a given flight day. As mentioned above, a median R$_{CO}$ value for each day is used as opposed to a more general, regression slope between flask CO ABL-FT differences and CO$_{2ff}$ because of low correlations between the two variables seasonally. We note that the numerator of R$_{CO}$ represents the enhancement in all CO sources (fossil, but also potentially biomass burning and oxidation of volatile organic compounds (VOCs)) as well as a very small amount of CO oxidative loss. Because all of these sources are "calibrated" relative to CO$_{2ff}$ (the denominator), it is not necessary to explicitly define non-fossil components of this ratio.

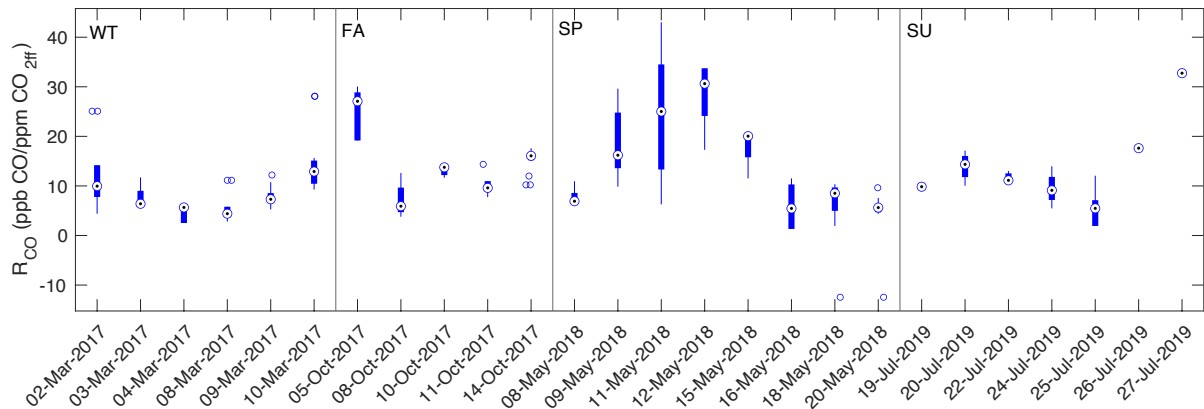

**Figure 6. Daily R$_{CO}$ calculated from discrete flask samples during winter (WT, 2017), fall (FA, 2017), spring (SP, 2018), and summer (SU, 2019). Note that, unlike Figures 4 and 5, flight data is shown in chronological order. Median R$_{CO}$ for each research flight is shown as a bullet point, with the interquartile (IQ) range shown as the thick bar line. Outliers greater than 1.5 times the IQ range are shown as separate open circles, and maximum extreme values as "whiskers" that do not qualify as outliers (thin lines) extending from the IQ ranges. Flask data shown are combined for B-200 and C-130 aircraft.**

Figure 6 indicates that the median R$_{CO}$ calculated for all ACT missions (10.03 ± 8.15 ΔCO per ppm CO$_{2ff}$; 68% CI) is slightly higher than in urban studies (Turnbull et al., 2011a; Miller et al., 2012; Turnbull et al., 2014). Day-to-day or even diurnal R$_{CO}$ is also largely variable, which could be a result of variability in CO$_{2ff}$, or in measured background levels (which are represented

by a single daily value). Spatial differences in VOC oxidation could play a minor role in influencing the variability of R$_{CO}$. CO additions from VOCs throughout the ACT Mid-Atlantic region will render R$_{CO}$ higher. However, despite the expectation that greater oxidation of VOCs will produce more CO during the summer (Vimont et al., 2017), we see the highest average R$_{CO}$ values during spring. R$_{CO}$ variability in both the fall and spring campaigns suggest that there could be other influential mechanisms occurring, yet DiGangi et al. (2021) found that biomass burning influence on air sampled during ACT were

negligible in the Mid-Atlantic region, ruling out the possibility that biomass burning events contribute to the high variability of R$_{CO}$ calculated here. In general, it is more likely that low CO$_{2ff}$ signals with high relative uncertainties are creating abnormally high R$_{CO}$ values using this median method in this work, which has also been found and discussed in great detail in Maier et al., (2024b).

As mentioned above, we calculate CO$_{2ff}$' only for days when flask measurements were analyzed for $\Delta^{14}$CO$_2$. When comparing CO$_{2ff}$' with flask-based CO$_{2ff}$, the mean bias is 0.7±2.1 ppm (1σ) (Fig. 7). The relatively large standard deviation in Fig. 7 indicates that flask samples collected within a single research flight were not always representative of the entire domain surveyed on that day by continuous analyzers. Point sources of CO$_2$ from power plants that do not emit CO captured by

continuous measurements could potentially skew this comparison alongside other non-fossil fuel CO sources, but the small
bias between the two suggests that $CO_{2ff}$' is generally in good agreement with flask $CO_{2ff}$.

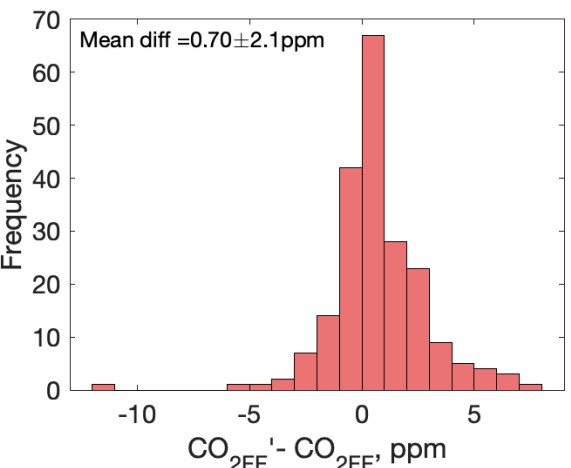

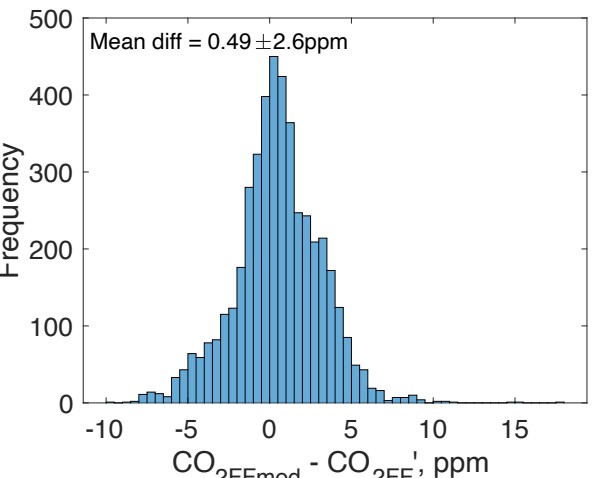

**Figure 7. Histogram of in situ ($CO_{2ff}$') minus flask-derived ($CO_{2ff}$) fossil fuel $CO_2$ for total-campaign corresponding measurement times (winter 2017 through summer 2019) in the Mid-Atlantic region. Here, $CO_{2ff}$' is calculated for corresponding flask sample times. By definition of $CO_{2ff}$ and $CO_{2ff}$' above, this comparison indicates ABL sample comparisons only.**

**Figure 8. Histogram of PSUWRF $CO_{2ffmod} - CO_{2ff}$' for all flask $\Delta^{14}CO_2$ sample days between winter 2017 and summer 2019 in the Mid-Atlantic region.**

### 3.3.2 Comparison of $CO_{2ff}$' and $CO_{2ffmod}$

Figure 8 shows the average difference between $CO_{2ffmod}$ computed along ACT flight tracks and ABL $CO_{2ff}$'. Several research flight days indicate rough agreement between $CO_{2ffmod}$ and $CO_{2ff}$' within a mean of ~0.49±2.6 ppm (1σ), but higher differences
outside of average $CO_{2ff}$' uncertainty bounds are seen in Fig. 8 along with a slight skewness. The mapped extent of this positive bias in $CO_{2ffmod}$ is visualized in Figure 9 for fall 2017, spring 2018 and summer 2019 campaigns, while the winter 2017 campaign model bias in $CO_{2ffmod}$ is slightly negative.

While there are logistical advantages to using this "pseudo-$CO_{2ff}$" method from a $\Delta^{14}CO_2$ measurement capacity standpoint, this product has a higher uncertainty (4.96 ppm) associated with large intra-day background variability and assumptions in the value of $R_{CO}$. For this reason, we caution the overgeneralization of the use of $CO_{2ff}$ as a transport tracer with this dataset and have chosen July 24, 2019 as an example of where positively biased $CO_{2ffmod}$ can be evaluated using $CO_{2ff}'$. Here, a series of B-200 flights was conducted downwind of three major eastern U.S. cities (New York City, NY; Philadelphia, PA; and Washington, DC). ABL flask sampling on each flight leg informed the derivation of $CO_{2ff}'$, and both estimates are in rough agreement as seen in Fig. 10. This individual flight day had higher fossil fuel $CO_2$ signals and lower variability in background CO measurements, with a lower $CO_{2ff}'$ uncertainty of ~2 ppm relative to the campaign-wide average.

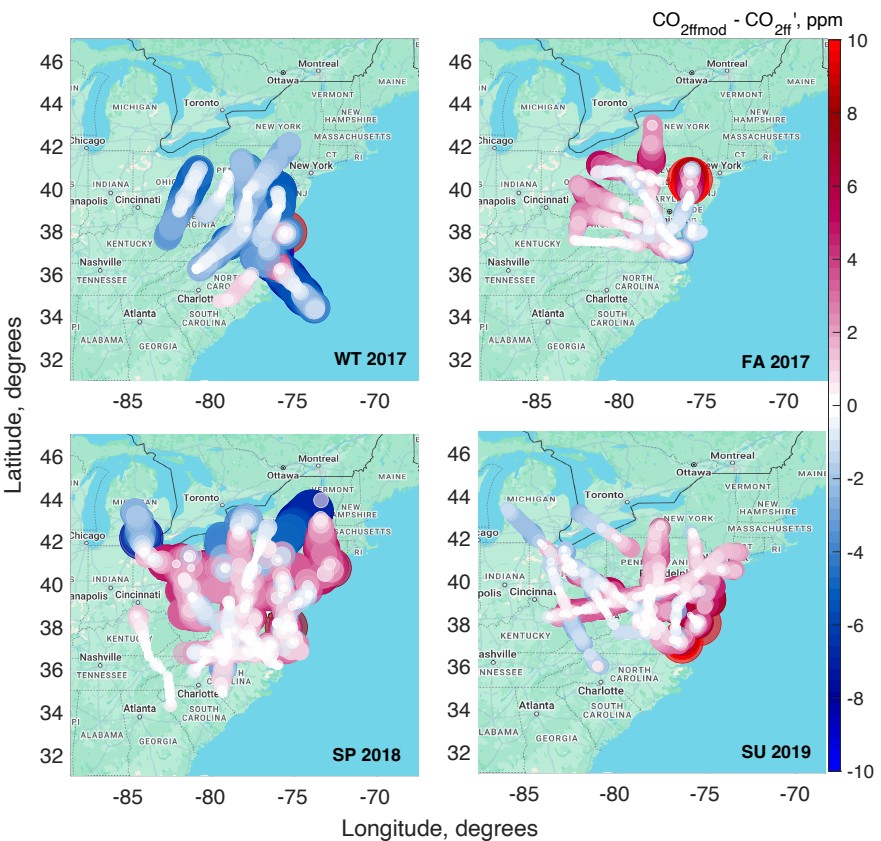

**Figure 9. Mapped modeled versus $CO_{2ff}'$ for flask $\Delta^{14}CO_2$ sample flight days for a) winter (WT) 2017, b) fall (FA) 2017, c) spring (SP) 2018 and d) summer (SU) 2019 campaigns. Points are sized by the model-data differences. Plot maps are produced using Google Maps API (©Google Maps).**

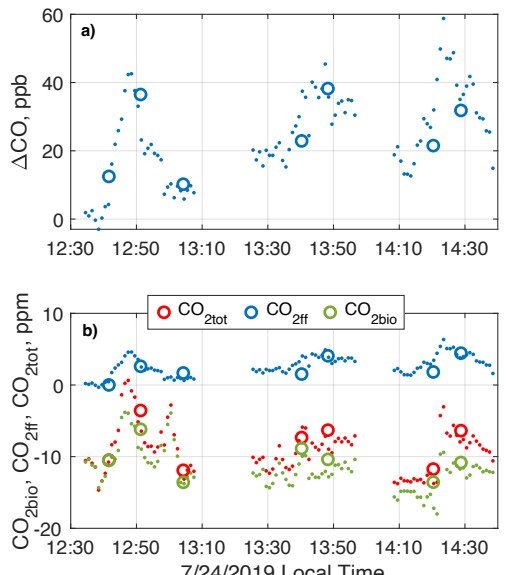

Figure 10. In situ (points) and flask (circles) observations for the B-200 aircraft on July 24, 2019. (a) CO ABL-FT differences (ΔCO) calculated from flasks and in situ continuous measurements. ΔCO from flasks is used to calculate $R_{CO}$ and thus $CO_{2ff}'$ using Eq. 5 using in situ measurements. The average difference in ΔCO (in situ – flask) is 0.6±8.7 ppb. (b) $CO_{2bio}$, $CO_{2ff}'$, and $CO_{2tot}$ calculated from both in situ high-frequency data (points) and from flask measurements (circles) on July 24, 2019 showing good agreement. Average differences in $CO_{2bio}$, $CO_{2ff}'$, and $CO_{2tot}$ (in situ – flask) are -1.6 ± 1.2 ppm, 1.2 ± 1.3 ppm and 0.4 ± 1.2ppm, respectively.

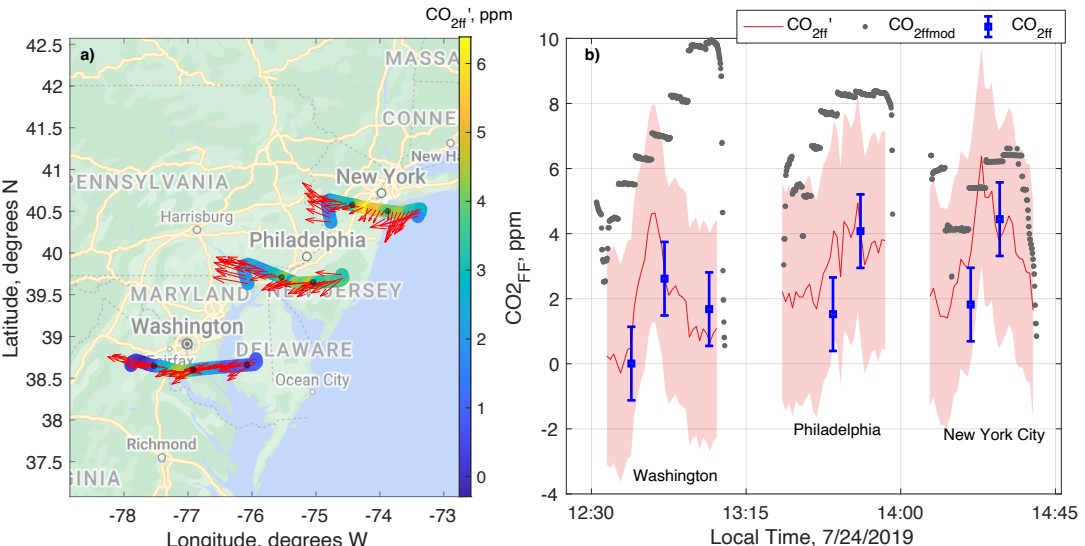

Figure 11. a) Pseudo-$CO_{2ff}$ ($CO_{2ff}'$) calculated along the ACT B-200 ABL flight track on July 24, 2019 (plot produced using Google Maps API, ©Google Maps). Flight tracks are colored by $CO_{2ff}'$ which proceed northerly from Washington, DC to New York City, NY. Black points indicate flask sample locations. Red arrows indicate the wind direction as calculated from aircraft flight data. b) Fossil fuel $CO_2$ calculated from in situ measurements ($CO_{2ff}'$), from flask $\Delta^{14}CO_2$ ($CO_{2ff}$) and from PSUWRF using CT fossil fuel emissions ($CO_{2ffmod}$). Discretization in $CO_{2ffmod}$ is due to the native 27-km resolution of PSUWRF. Error shading on

**CO$_{2FF}$' is the derived 1σ uncertainty on the CO$_{2ff}$' calculation for July 24, 2019 and error bars on flask CO$_{2ff}$ reflect a 1σ uncertainty.**

Figure 11 shows calculated CO$_{2ff}$' along three ABL flight legs with large positive fossil fuel CO$_2$ downwind of these urban areas. Comparing modeled (CO$_{2ffmod}$), calculated (CO$_{2ff}$'), and flask-derived (CO$_{2ff}$) CO$_{2ff}$, we find that CO$_{2ff}$' plumes observed
downwind of cities do not always align with CO$_{2ffmod}$ throughout the research flight. Significant differences are observed between CO$_{2ffmod}$ and CO$_{2ff}$' magnitudes, particularly during the Washington, D.C. flight leg of up to 60% beyond uncertainty bounds, with the urban emissions plume misaligned (Fig. 11). CO$_{2ffmod}$ is consistently higher than CO$_{2ff}$' in the Philadelphia flight leg as well, with general agreement in the plume variability. Finally, we note that the PSUWRF model simulates CO$_{2ff}$ in the New York City flight leg relatively well with respect to flask CO$_{2ff}$.


Current uncertainties of CO$_2$ fossil fuel fluxes, based on differences among various emissions products, are less than or equal to 20% at the regional scale (Gurney et al., 2020), and this is corroborated in the PSUWRF model, where Feng et al., (2019a) note that the uncertainty in fossil fuel fluxes on a daily average time scale is approximately 20%. This uncertainty may increase at the hourly timescale and certainly within smaller, urban-scale domains (Gately and Hutyra, 2017). We investigate several
reasons for the 60% overestimation in simulated CO$_{2ff}$ beyond the larger calculated CO$_{2ff}$' uncertainty bounds and the misalignment of the Washington, D.C. CO$_{2ffmod}$ "plume" relative to CO$_{2ff}$' could occur. First, this discrepancy cannot be explained by model deviations in key meteorological variables such as wind speed, wind direction or atmospheric boundary layer depth relative to observations. The PSUWRF modeled wind speeds and directions compare well to in situ wind speed and directions along the B-200 flight track to within 1 m s$^{-1}$ and 10 degrees, respectively. ABL depths derived from B-200
potential temperature soundings flown at the endpoints of each urban transect in Fig. 11 also compare well with ABL depth estimated from PSUWRF potential temperature to within 300 m on average. Second, emissions uncertainties within PSUWRF could result in observed differences between CO$_{2ff}$' and CO$_{2ffmod}$ of this magnitude. One hypothesis is that, if multiple fossil fuel emissions datasets are run independently in PSUWRF and the ensemble of these simulations is narrow, then more confidence can be gained in that the differences in Fig. 11 are not due to errors in fluxes, but rather in model transport.
PSUWRF simulations between 2017-2018 indicate that independently-run fossil fuel emissions fields with differing spatial and temporal resolution (i.e. ODIAC and Miller fields) result in small CO$_{2ffmod}$ differences that are within 1.5-2 ppm on average. While these fossil fuel emissions products were not run separately in the PSUWRF model beyond 2018, a 2 ppm CO$_{2ffmod}$ variability is substantially smaller than that shown in Fig. 11 downwind of Washington and Philadelphia (3-6 ppm CO$_{2ff}$). It is possible that the model vertical transport parameterizations are erroneous earlier in the day, which could produce
errors later in the day in CO$_{2ff}$ accumulation and venting relative to that observed downwind of Washington, D.C., and even Philadelphia, PA. Therefore, while our current results suggest that model transport, rather than fluxes are erroneous on this particular flight, the results from this case study are important to document for future model-observation comparisons. More intensive work beyond the scope of this work would be needed to verify our hypothesis, which could involve model runs with

a number of both transport and flux variants to discern model variability with different, realistic ensemble members. Within this type of study, observed and modeled values for additional urban campaign data should also be compared.

While this example does not represent the average $CO_{2ffmod}$ bias with respect to $CO_{2ff}$' during ACT, it illustrates the utility of $CO_{2ff}$ as a model transport tracer. In the above example, the high-frequency $CO_{2ff}$' product is able to easily highlight potential modeled errors in the representation of diurnal $CO_{2ff}$ plume variability. As discussed in Miller et al. (2020), deriving $CO_{2ff}$' in

this way, using information gleaned from a small subset of $\Delta^{14}CO_2$ flask sample measurements, can provide a means for determining the fossil fuel and biogenic components of $CO_{2tot}$ for regional or city-scale studies. Despite higher average uncertainties than flask-based methods, comparisons to models might allow for examination and improvements upon gross model errors for more accurate $CO_2$ mole fraction estimation and process-based studies. As mentioned above, differentiating between regional model transport and flux errors might be improved upon in the future by using an ensemble of both transport

and flux variants. As ACT $\Delta^{14}CO_2$ flask sampling was generally aimed at capturing broad-scale biogenic $CO_2$ features across the eastern U.S., more targeted flask sampling in urban areas with larger $CO_{2ff}$ signals could reduce $R_{CO}$ and $CO_{2ff}$' uncertainties, which would allow for more robust $\Delta^{14}CO_2$-based model verification using comparatively fewer flask samples. At more rural sites, a greater number of flask samples may be needed to robustly calculate $CO_{2ff}$' (Maier et al., 2024b).

## 4 Conclusions

The ACT-America mission, while focused on the improvement of the accuracy of regional carbon flux estimates, also provided the opportunity for a dense, seasonally-diverse dataset of atmospheric $\Delta^{14}CO_2$ measurements from flasks sampled throughout the Mid-Atlantic U.S. and the capability to disaggregate total $CO_2$ into biogenic and fossil fuel components, which is critical for regional $CO_2$ source attribution. Seasonal campaigns occurring between 2017 and 2019 indicated that biogenic $CO_2$ exchange in the Mid-Atlantic was the primary driver of $CO_2$ boundary layer variability in this region as expected, while $CO_{2ff}$

remained relatively constant during the ACT mission. Consistent OCS uptake in all seasons was observed that correlated well with negative $CO_{2bio}$, confirming $CO_2$ uptake by photosynthesis throughout the ACT campaigns, though instances were seen where OCS uptake was clearly decoupled from gross primary productivity during the fall. With the broad nature of flask sampling during ACT, the signal to noise (measurement precision) ratio in $\Delta^{14}CO_2$ data was low and campaigns investigating strictly urban $CO_2$ signals can better highlight the utility that routine observations of $\Delta^{14}CO_2$ can provide, including critical

information for stakeholders in assessing carbon reduction strategies on regional to sub-regional scales. Several studies have shown the value of incorporating $\Delta^{14}CO_2$ alongside $CO_2$ measurements as an improved model constraint on regional $CO_2$ fluxes and future work will require a continued effort to ensure routine $\Delta^{14}CO_2$ measurements throughout the NOAA GGGRN and other networks. Here, we have used flask $\Delta^{14}CO_2$ samples, taken alongside continuous in situ CO measurements, to provide a high-frequency "pseudo-$CO_{2ff}$" product ($CO_{2ff}$') using the relationship between in situ CO ABL-FT differences and flask-

derived $CO_{2ff}$. This product was used to evaluate modeled $CO_{2ff}$ at a higher temporal resolution than discrete flasks can provide, and to illuminate potential errors in model transport at the regional scale. Although the ACT dataset and analysis therein is limited by higher $CO_{2ff}$' uncertainties, and to days where $\Delta^{14}CO_2$ signals are higher, this work highlights the value of such a product in future campaigns and measurement networks as a model evaluation tool.

**Data Availability**

Observational in situ and flask data from the NASA ACT-America mission used for the analysis in this study are publicly archived at the Oak Ridge National Laboratory repository at https://doi.org/10.3334/ORNLDAAC/1593 (Davis et al., 2018) in addition to PSUWRF output for the ACT-America campaign at https://doi.org/10.3334/ORNLDAAC/1884 (Feng et al., 2021).. Individual flask data with $^{14}CO_2$ measurements from ACT-America can be found at
(https://doi.org/10.3334/ORNLDAAC/1575). All data are synced to meteorological and location data onboard each aircraft and available for downloading and merging (https://doi.org/10.3334/ORNLDAAC/1574). GGGRN $\Delta^{14}CO_2$ and CO flask data used to supplement this study from NWR, NHA, and CMA are available at https://doi.org/10.15138/87ny-6277.  The NOAA HYSPLIT model is publicly available via https://www.ready.noaa.gov/HYSPLIT.php for registered and unregistered versions.

**Author Contributions**

BCB led the investigation and curated the data alongside JPD, YC, SL and CW. CS, KD, JM, JPD and SL assisted with acquiring funding for this work. BB, JM, and SL conceptualized the analysis and contributed to writing the original draft of this manuscript. SF and TL developed and contributed model analysis and output. All authors contributed to reviewing and editing the manuscript.

**Competing Interests**

The authors declare that they have no conflict of interest.

**Acknowledgements**

They acknowledge the collaboration between NOAA Global Monitoring Laboratory (GML) and its cooperative partners,
including its collaboration with the Cooperative Institute for Research in Environmental Sciences (CIRES) and facilities available to the Institute for Arctic and Alpine Research (INSTAAR). The authors thank Jocelyn Turnbull and Stephen Montzka for helpful discussions of data analysis methods and to Zachary Barkley for conducting PSUWRF model-observation comparisons of meteorological variables. We also acknowledge the critical work of NOAA/GML personnel Patricia Lang, Andrew Crotwell, Monica Madronich, Eric Moglia, Benjamin R. Miller, Duane Kitzis and Molly Crotwell with the
measurement of flask samples and calibration of measurements, and NOAA collaborative partners throughout the GGGRN with the collection of aircraft flask samples at CMA, NHA and NWR. We acknowledge that this work would not have been

made possible without significant help from the ACT-America science team, pilots, and NASA management support. Partial funding for this work was provided to the University of Colorado/CIRES from NASA grant NNX15AJ06G and the NOAA Cooperative Agreement with CIRES, NA17OAR4320101 and by NASA's Earth Science Division grant NNX15AG76G to

Penn State. S. Feng at PNNL is supported by the NASA Carbon Monitoring Program (Grant number: 80HQTR21T0069). The Pacific Northwest National Laboratory is operated by Battelle Memorial Institute under contract DE-A05-76RL0 1830.

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
