# Peer review of "Mid-Atlantic U.S. observations of radiocarbon in CO2: fossil and biogenic source partitioning and model evaluation"

_EGUsphere, 2025_

## Referee Comment (RC1)

**Review of Baier et al. (2025), „Mid-Atlantic U.S. observations of radiocarbon in CO$_2$: fossil and biogenic source partitioning and model evaluation"**

This study investigates and analyses atmospheric observations made during four aircraft campaigns over the northwestern U.S. in different seasons between 2017 and 2019, with a particular focus on the collected $\Delta^{14}CO_2$ flask samples. In a first part, the $\Delta^{14}CO_2$ flask observations were used to interpret the variability of the $CO_2$ enhancements in the boundary layer by separating the fossil from the biogenic contributions. The second part of the study focuses on model evaluation and provides a comparison between modelled and observed $CO_{2ff}$ concentrations along the flight paths, the latter derived from high-frequency CO measurements and $R_{CO}$ ratios. In a case study of a single afternoon flight, the authors show how the detection of urban $CO_{2ff}$ plumes could help to identify transport model deficits.

In my opinion, this is a relevant study that nicely demonstrates how $\Delta^{14}CO_2$ flask observations from aircraft missions can be used to estimate $CO_{2ff}$ enhancements and to interpret $CO_2$ signals in the boundary layer over large continental regions. The results are presented in a clear way and the associated uncertainties are carefully calculated. However, what I'm missing a bit is a concluding discussion of what can be learned from the findings of the case study for potential future aircraft missions aimed at evaluating transport model performance, e.g. in different regions. Should such missions focus on urban regions so that the uncertainty in the pseudo $CO_{2ff}$ is small enough for model evaluation purposes? Furthermore, since it is difficult to separate the effects of transport model errors and potential flux biases on modelled $CO_{2ff}$ concentrations, should only plumes from regions with well-known $CO_{2ff}$ fluxes be sampled? Regarding the flask sampling approach, could a (realistic) increase in the $\Delta^{14}CO_2$ data collected during a flight (or a different ratio between ABL and FT flasks) significantly reduce the uncertainty of the pseudo $CO_{2ff}$ record used for model evaluation? If you could elaborate a little on these (exemplary) points, I think it would add to the relevance and applicability of this study.

Overall, this study fits well within the scope of ACP and I recommend it for publication after addressing my (minor) comments below.

**Specific comments:**

p. 1, l. 17: Please explain the abbreviation "ACT".

p. 2, l. 64: "$CO_{2ff}$" has not yet been introduced.

p. 3, l. 68: It seems that the study from Graven et al. (2018) is not based on "aircraft $\Delta^{14}CO_2$ observations". They used $\Delta^{14}CO_2$ observations from nine measurement sites in California.

p. 3, l. 75: From what is given in Sect. 2.1., it seems that - apart from the first ACT campaign, which you have excluded for this analysis anyway - the campaigns lasted 2-3 weeks rather than 6 weeks. Please clarify.

p. 3, l. 82: Briefly saying that/why OCS can be used as a tracer for photosynthesis might be helpful here.

p. 3, l. 84: Please cite the final revised version of this manuscript instead:
Maier, F., Rödenbeck, C., Levin, I., Gerbig, C., Gachkivskyi, M., and Hammer, S.: Potential of 14C-based vs. $\Delta$CO-based $\Delta ffCO_2$ observations to estimate urban fossil fuel $CO_2$ ($ffCO_2$) emissions, Atmos. Chem. Phys., 24, 8183–8203, https://doi.org/10.5194/acp-24-8183-2024, 2024.

p. 4, l. 111: How long was the flask filling time, i.e. for what time interval are the flask samples representative?

p. 5, l. 137: What about $CO_2$ contributions from the ocean? Fig. 1 indicates that there may be an oceanic influence in spring and fall.

p. 7, l. 196ff: It would be interesting to have the contribution of the $CO_{2corr}$ also in relative numbers of the total $CO_{2ff}$ signal.

p. 7, l. 198-200: To be consistent with what is written in l. 197, change "$CO_2$" into "$CO_{2ff}$".

p. 7, l. 200: Perhaps you could add that the lack of a clear seasonal cycle in the nuclear $CO_{2corr}$ is consistent with the fact that you assumed constant nuclear emissions.

p. 7, l. 206: Please explain what PSUWRF stands for. What is the vertical resolution of the Eulerian model?

p. 7, l. 216-219:  This is not clear to me. Why did you create a single flux product only for model runs in 2019? What are you using for 2017-2018? Can this lead to inconsistencies in the $CO_{2ff}$ simulations for the different years? Please clarify.

p. 8, l. 239-241: It is unclear to me how you've calculated $R_{CO}$. From what is written in this paragraph, I would assume that you've used the median CO enhancement from the continuous measurements and not the CO enhancements from the flask measurements to calculate $R_{CO}$. Is this true? Is the argument, that the variability of the CO enhancement within one flight is larger than the variability of the $CO_{2ff}$ enhancement. And that the CO enhancements of the flasks are therefore less representative for the whole flight than the $CO_{2ff}$ from the flasks, which is why you've used the median CO enhancement from the continuous measurements and the $CO_{2ff}$ from the flasks to calculate $R_{CO}$? However, in contrast to that is the caption of Fig. 10: "$\Delta$CO from flasks is used to calculate $R_{CO}$...". Please motivate and explain the calculation of $R_{CO}$. It would be helpful to have an additional equation for the calculation of $R_{CO}$.

Fig. 1: Change "white" -> "black" in the caption.

p. 10, l. 281ff: From Fig. 2b, it appears that the winter 2017 FT samples are not "slightly higher" than the 3-month averaged GGGRN measurements (only the fall and spring samples seem to be higher).

Fig. 2: Does the shaded area of the 3-month averaged GGGRN data indicate the 1-sigma standard deviation?

p. 11, l. 294-295: One could add here that negative $CO_{2ff}$ values could also be explained by an inappropriate $\Delta^{14}CO_2$ background or by underestimated nuclear/bio masking of $CO_{2ff}$.

p. 11, l. 295ff: Have those studies also used the FT as a $\Delta^{14}CO_2$ background to calculate the $CO_{2ff}$ concentrations?

p. 13, l. 327: Can part of the weak correlations between $CO_{2tot}$ and $CO_{2ff}$ be explained by the small $CO_{2ff}$ signals?

p. 13, l. 330: The spring campaign took place in May, not in March-April, right?

p. 14, l. 332: The fraction of $CO_{2bio}<0$ is similar in fall and winter (8% and 9%, respectively). Could the $CO_{2bio}<0$ data in winter be explained by the fact that the winter campaign took place in March when biosphere is already starting to be active or is it due to observational uncertainties (i.e. is the small fraction of $CO_{2bio}<0$ significant in winter)?

p. 14, l. 341-342: Do you mean: "when combining all seasonal ACT data except fall"?

Fig. 6: Maybe you could briefly mention in the caption of Fig. 6 that you have changed the ordering of the seasons (compared to the previous plots in Fig. 4 and 5).

p. 16, l. 377ff: Another possibility for the large variability in $R_{CO}$ are measurement uncertainties. In Maier et al. (2024b, see the appendix A1 of the study), we have seen that the large relative uncertainty of small $CO_{2ff}$ signals can lead to spuriously high $R_{CO}$, even in the absence of natural CO sources:
Maier, F., Levin, I., Conil, S., Gachkivskyi, M., Denier van der Gon, H., and Hammer, S.: Uncertainty in continuous $\Delta CO$-based $\Delta ffCO_2$ estimates derived from $^{14}C$ flask and bottom-up $\Delta CO / \Delta ffCO_2$ ratios, Atmos. Chem. Phys., 24, 8205–8223, https://doi.org/10.5194/acp-24-8205-2024, 2024b.

Fig. 8: Is the distribution of the model-data differences in hourly resolution?

p. 20, l. 414: Is the 20% uncertainty in the fossil fuel $CO_2$ fluxes also appropriate for the temporal scale of individual hours of the day? I'm wondering if an inappropriate diurnal profile in the $CO_{2ff}$ fluxes could contribute to the observed model-data differences? For a better interpretation of the results, it would be helpful to show the time axis in Fig. 11 in local time instead of UTC, or to indicate in the caption of Fig. 11 what 16:30 UTC is in local time.

p. 20, l. 415ff: Could a slight misalignment in the modelled wind direction be the reason for the temporal shift in the maximum of modelled $CO_{2ff}$ in Washington?

**Technical corrections:**

p. 2, l. 43: Delete ", however" or connect it with the next sentence.

p. 8, l. 225: "Maier et al., 2023" -> "Maier et al., 2024"

p. 10, l. 288: Delete „or"?

p. 12, l. 314: "four years" -> "three years"?

p. 16, l. 381: „Spring" -> „spring"

Fig. 11: „Psuedo" -> „Pseudo" (in the caption)

p. 20, l. 435: "Maier et al., 2023" -> "Maier et al., 2024"

---

## Author Response (AR1)

**General comment on the ACT campaign and $R_{CO}$ to both reviewers:**

The ACT-America mission was directed toward improving the accuracy of regional greenhouse gas flux estimates, specifically through refining our understanding and implementation of $CO_2$ and methane ($CH_4$) transport, fluxes and flux uncertainties and background levels in inverse models (Davis et al., 2021). Most research flights spanned large rural areas within the eastern U.S. targeting the transport of $CO_2$ across weather fronts as well as biogenic $CO_2$ signatures. Radiocarbon samples were included in the overall ACT-America flask sampling strategy (Baier et. al., 2020) for the Mid-Atlantic U.S. only due to this region having the largest potential $CO_{2ff}$ signatures; however, the signal to noise ($\Delta^{14}CO_2$ precision) – even in this region – was low due to the overall ACT-America flight characteristics, somewhat limiting the extent of this analysis. This study examines, to the best of our ability, $CO_{2ff}$ signals from these flask sample data, alongside comparisons to the WRF model.

We thank both reviewers for their thorough examination of $R_{CO}$ variability. The largest source of this variability is the low $CO_{2ff}$ signal with relatively high measurement uncertainty. While we considered using regression to determine $R_{CO}$, it was difficult to perform regressions due to the high scatter and low $R^2$ values (ranging from 0.08-0.33) between flask $\Delta CO$ and $CO_{2ff}$ observed during each ACT seasonal campaign – again, due to the airborne sampling strategy. Rather than utilizing weak regressions, and while there are uncertainties associated with both methods for this dataset (as mentioned in Maier et al., (2024a)), we ultimately decided that the median method for this work was the most robust way to calculate $R_{CO}$ values. $R_{CO}$ is calculated for each flight day as opposed to relying on a single value for each ACT seasonal campaign, and this variability is a large source of the overall uncertainty in $CO_{2ff}$'.

**Reviewer 1**

**Review of Baier et al. (2025), „Mid-Atlantic U.S. observations of radiocarbon in $CO_2$: fossil and biogenic source partitioning and model evaluation"**

This study investigates and analyses atmospheric observations made during four aircraft campaigns over the northwestern U.S. in different seasons between 2017 and 2019, with a particular focus on the collected $\Delta^{14}CO_2$ flask samples. In a first part, the $\Delta^{14}CO_2$ flask observations were used to interpret the variability of the $CO_2$ enhancements in the boundary layer by separating the fossil from the biogenic contributions. The second part of the study focuses on model evaluation and provides a comparison between modelled and observed $CO_{2ff}$ concentrations along the flight paths, the latter derived from high-frequency CO measurements and $R_{CO}$ ratios. In a case study of a single afternoon flight, the authors show how the detection of urban $CO_{2ff}$ plumes could help to identify transport model deficits.

Line numbers refer to "Simple" and/or "No" Markup version of the manuscript.

We thank the reviewer for taking the time to conduct a thorough review and for their insightful comments and suggestions for improving this manuscript. Please see author responses below in blue that address these comments.

In my opinion, this is a relevant study that nicely demonstrates how $\Delta^{14}CO2$ flask observations from aircraft missions can be used to estimate $CO_{2ff}$ enhancements and to interpret $CO_2$ signals in the boundary layer over large continental regions. The results are presented in a clear way and the associated uncertainties are carefully calculated. However, what I'm missing a bit is a concluding discussion of what can be learned from the findings of the case study for potential future aircraft missions aimed at evaluating transport model performance, e.g. in different regions. Should such missions focus on urban regions so that the uncertainty in the pseudo $CO_{2ff}$ is small enough for model evaluation purposes? Furthermore, since it is difficult to separate the effects of transport model errors and potential flux biases

on modelled $CO_{2ff}$ concentrations, should only plumes from regions with well-known $CO_{2ff}$ fluxes be sampled? Regarding the flask sampling approach, could a (realistic) increase in the $\Delta^{14}CO_2$ data collected during a flight (or a different ratio between ABL and FT flasks) significantly reduce the uncertainty of the pseudo $CO_{2ff}$ record used for model evaluation? If you could elaborate a little on these (exemplary) points, I think it would add to the relevance and applicability of this study.

In L482-493, we write (in reference to a concluding discussion of the model vs. observations):
"While this example does not represent the average $CO_{2ffmod}$ bias with respect to $CO_{2ff}$' during ACT, it illustrates the utility of $CO_{2ff}$ as a model transport tracer. In the above example, the high-frequency $CO_{2ff}$' product is able to easily highlight potential modeled errors in the representation of diurnal $CO_{2ff}$ plume variability. As discussed in Miller et al. (2020), deriving $CO_{2ff}$' in this way, using information gleaned from a small subset of $\Delta^{14}CO_2$ flask sample measurements, can provide a means for determining the fossil fuel and biogenic components of $CO_{2tot}$ for regional or city-scale studies. Despite higher average uncertainties than flask-based methods, comparisons to models might allow for examination and improvements upon gross model errors for more accurate $CO_2$ mole fraction estimation and process-based studies. As mentioned above, differentiating between regional model transport and flux errors might be improved upon in the future by using an ensemble of both transport and flux variants. As ACT $\Delta^{14}CO_2$ flask sampling was generally aimed at capturing broad-scale biogenic $CO_2$ features across the eastern U.S., more targeted flask sampling in urban areas with larger $CO_{2ff}$ signals could reduce $R_{CO}$ and $CO_{2ff}$' uncertainties, which would allow for more robust $\Delta^{14}CO_2$-based model verification using comparatively fewer flask samples. At more rural sites, a greater number of flask samples may be needed to robustly calculate $CO_{2ff}$' (Maier et al., 2024b). "

While the overall precision in $\Delta^{14}CO_2$ measurements is remarkable, increasing this to better than 2‰ would also reduce the uncertainty of the pseudo $CO_{2ff}$ record.

Overall, this study fits well within the scope of ACP and I recommend it for publication after addressing my (minor) comments below.

Specific comments:

p. 1, l. 17: Please explain the abbreviation "ACT".

Fixed in L18, thank you.

p. 2, l. 64: "$CO_{2ff}$" has not yet been introduced.

Fixed in L46, thank you.

p. 3, l. 68: It seems that the study from Graven et al. (2018) is not based on "aircraft $\Delta^{14}CO_2$ observations". They used $\Delta^{14}CO_2$ observations from nine measurement sites in California.

This is correct. We have deleted this reference in L70 and replaced with Basu et al., 2020 which is also a relevant citation.

p. 3, l. 75: From what is given in Sect. 2.1., it seems that - apart from the first ACT campaign, which you have excluded for this analysis anyway - the campaigns lasted 2-3 weeks rather than 6 weeks. Please clarify.

The length of the campaigns is listed in Section 2.1 (2-3 weeks), so we have omitted the incorrect duration of each campaign in this more generalized sentence (L83) to read "During this mission, five seasonal research flight campaigns were conducted between… in the eastern U.S. carbon monitoring network."

p. 3, l. 82: Briefly saying that/why OCS can be used as a tracer for photosynthesis might be helpful here.

We have expanded the introduction L73-79 to include more of an explanation of how OCS can be helpful:

"As biogenic $CO_2$ emissions and sinks – even within cities – are non-negligible, and ignoring these signals could potentially bias fossil fuel $CO_2$ emission inventories, $\Delta^{14}CO_2$ measurements play an important role in determining biogenically-driven ($CO_2$ photosynthesis and respiration) emissions or sinks (Levin et al., 2003; Lopez et al., 2013; Turnbull et al., 2015; Miller et al., 2020). One tracer that could aid in the understanding of $CO_2$ uptake processes alongside $\Delta^{14}CO_2$ is carbonyl sulfide (OCS), which is taken up by plants but not respired (Montzka et al., 2007; Campbell et al., 2008)."

p. 3, l. 84: Please cite the final revised version of this manuscript instead:
Maier, F., Rödenbeck, C., Levin, I., Gerbig, C., Gachkivskyi, M., and Hammer, S.: Potential of 14C-based vs. ΔCO-based ΔffCO$_2$ observations to estimate urban fossil fuel $CO_2$ (ffCO$_2$) emissions, Atmos. Chem. Phys., 24, 8183–8203, https://doi.org/10.5194/acp-24-8183-2024, 2024.

This citation has been revised, thank you.

p. 4, l. 111: How long was the flask filling time, i.e. for what time interval are the flask samples representative?

L118 was added to state that "Flask filling times varied between two (ABL) to four minutes (FT) depending on the altitude of the aircraft."

p. 5, l. 137: What about $CO_2$ contributions from the ocean? Fig. 1 indicates that there may be an oceanic influence in spring and fall.

L158 states that "We note that, an oceanic term is sometimes written in Equation (1), but we assume that the dominant oceanic $CO_2$ influence is encompassed in the background (*bg)* term."

With this being said, Figure 1 does indicate some oceanic influence in the spring and fall, but this signal is small relative to the total footprint for each season. L203-204 further explain, "Referring to Eqns. (1) and (2), though there exists a footprint signature on flask samples during fall 2017 and spring 2018, this influence is relatively small."

p. 7, l. 196ff: It would be interesting to have the contribution of the $CO_{2corr}$ also in relative numbers of the total $CO_{2ff}$ signal.

In L216 we have added " In total, the average magnitude of $CO_{2corr}$  ($0.8 \pm 0.6$ ppm) calculated here to correct $CO_{2ff}$ in this work is roughly comparable that first described in Turnbull et al. (2006)."

p. 7, l. 198-200: To be consistent with what is written in l. 197, change "$CO_2$" into "$CO_{2ff}$".

This has been changed in L213-216 "The magnitude of $CO_{2corr}$ attributable to the terrestrial biosphere in the Mid-Atlantic U.S. is approximately $0.10 \pm 0.04$ ppm $CO_{2ff}$ ($1\sigma$) during the winter deployment and exhibits a seasonal cycle with a maximum during the summer of $0.70 \pm 0.28$ ppm $CO_{2ff}$ ($1\sigma$), indicating similar seasonal behavior as that calculated by Miller et al. (2012) for the Mid-Atlantic U.S."

p. 7, l. 200: Perhaps you could add that the lack of a clear seasonal cycle in the nuclear $CO_{2corr}$ is consistent with the fact that you assumed constant nuclear emissions.

Yes, this is a good suggestion and L210-213 has been changed to: " We note that the calculated magnitude of $CO_{2corr}$ attributable to nuclear reactor emissions in the Mid-Atlantic U.S. is 0.25 ± 0.44 ppm $CO_2$ and does not exhibit a seasonal pattern as it is assumed constant annually. Graven and Gruber (2011) have been used at the time that this publication was written, but updates to nuclear reactor fluxes have been published in Zazzeri et al. (2023)."

p. 7, l. 206: Please explain what PSUWRF stands for. What is the vertical resolution of the Eulerian model?

PSUWRF has been explained, with L219-221 reading "A separate, Eulerian 27 km horizontal resolution implementation of WRF-Chem version 3.6.1 run (Feng et al., 2021a,b) was implemented by the Pennsylvania State University (hereby called PSUWRF) with 50 vertical levels from the surface to 50 hPa, with 29 of these vertical levels in the lowermost 2 km."

p. 7, l. 216-219: This is not clear to me. Why did you create a single flux product only for model runs in 2019? What are you using for 2017-2018? Can this lead to inconsistencies in the $CO_{2ff}$ simulations for the different years? Please clarify.

PSUWRF runs use NOAA CarbonTracker fossil fuel $CO_2$ fluxes. Traditionally, CT uses an average of ODIAC and Miller emissions datasets to optimize the distribution of these emissions both spatially and temporally. These two flux products (having the same U.S. national totals, but different spatial and temporal distributions) were experimentally run separately in PSUWRF for 2016-2018 to investigate any differences between the two datasets, if any. However, $CO_{2ff}$ model output from these separate runs shows very good agreement between these two products when run independently, and so they were recombined in 2019 as in the CT model to save compute time and simplify future model runs. We take advantage of this experimental run of the flux products independently to reinforce the hypothesis that small differences between the two datasets are expected to lead to non-negligible inconsistencies in $CO_{2ff}$ simulations between 2016-2019. L237-242 has been modified to:

"As both datasets have very similar global and national fossil fuel emissions totals, but include differences spatial and temporal emissions distributions, an average of the two is used in CT to optimize the mapping of these emissions (Jacobson et al., 2020). In PSUWRF, ODIAC and Miller datasets were run separately for initial model runs between 2016-2018 to experimentally investigate potential differences between the two; however, only small differences within ~2 ppm $CO_{2ff}$ are seen between the two and are not expected to create inconsistencies in $CO_{2ff}$ simulated by PSUWRF between 2016-2019. As such, ODIAC and Miller are averaged to create a single flux product and to simplify model runs as in CT for 2019."

p. 8, l. 239-241: It is unclear to me how you've calculated $R_{CO}$. From what is written in this paragraph, I would assume that you've used the median CO enhancement from the continuous measurements and not the CO enhancements from the flask measurements to calculate $R_{CO}$. Is this true?

Yes. $R_{CO}$ is strictly calculated from flask measurements only. It is calculated by taking the median value of all available flask $\Delta CO$ and $\Delta^{14}CO_2$ collected each day, and propagating this median value to the *in situ* measurements to calculate $CO_{2ff}$'.

We have considered using regression to determine $R_{CO}$. However, it was difficult to perform regressions due to the high scatter and low $R^2$ values (ranging from 0.08-0.33) between flask $\Delta CO$ and $CO_{2ff}$ observed during each ACT seasonal campaign. Rather than utilizing weak regressions, and while there are uncertainties associated with both methods for this dataset (as mentioned in Maier et al., (2024a)), we ultimately decided that the median method for this work was the most robust way to calculate $R_{CO}$ values. We have calculated these for each flight day (as opposed to relying on a single value for each

ACT seasonal campaign). We have noted this in L266-271. We do propagate the uncertainty in $R_{CO}$ (from the width of a normally-distributed 68% confidence interval about average $R_{CO}$ values) into the overall uncertainty of $CO_{2ff}$'.

"A median $R_{CO}$ is used each day due to difficulty with respect to calculating seasonal regression slopes with low correlations (seasonal $R^2$ between 0.08-0.33) in the flask CO enhancement and $CO_{2ff}$ data. In calculating daily median $R_{CO}$, it may be possible to capture some spatial variability of research flight observations rather than ignoring this variability and rely on average seasonal values. However, we acknowledge in Section 3.3.1 that this methodology, given low signal to noise ($\Delta^{14}CO_2$ precision) in the ACT $CO_{2ff}$ data, could create anomalous variability in $R_{CO}$ and is one of the largest sources of uncertainty in this $CO_{2ff}$' calculation (as first presented in detail in Maier et al., 2024b) and a limitation to this analysis"

Is the argument, that the variability of the CO enhancement within one flight is larger than the variability of the $CO_{2ff}$ enhancement. And that the CO enhancements of the flasks are therefore less representative for the whole flight than the $CO_{2ff}$ from the flasks, which is why you've used the median CO enhancement from the continuous measurements and the $CO_{2ff}$ from the flasks to calculate $R_{CO}$? However, in contrast to that is the caption of Fig. 10: "ΔCO from flasks is used to calculate $R_{CO}$...". Please motivate and explain the calculation of $R_{CO}$. It would be helpful to have an additional equation for the calculation of $R_{CO}$.

$R_{CO}$ it is informed by flask measurements, and these measurements are not as spatially dense as in situ ones. Therefore, we state that we make the assumption that $R_{CO}$, as a daily median ratio, is representative of the entire flight region sampled, and we do not calculate $CO_{2ff}$' for days when flask $\Delta^{14}CO_2$ is not available to inform $CO_{2ff}$'.

We have adjusted L254 to read:

"We calculate $CO_{2ff}$' using Equation (5) by applying a median ratio of CO to $CO_{2ff}$ ($R_{CO}$) calculated strictly from flask samples collected each day (typically from 5-12 flask samples), and apply this value to the in situ CO time series collected on that day's ACT research flight:

$$CO_{2ff}' = \frac{CO_{obs}' - \overline{CO'_{bg}}}{R_{CO}}.$$  (5)

Here, $CO_{obs}'$ are 0.2 Hz ABL (below 1500 m ASL) in situ CO dry mole fraction observations and $\overline{CO'_{bg}}$ is the mean daily CO mole fraction observed in the FT (above 4000 m ASL) for each flight day, and

$$R_{CO} = \frac{[CO_{obs,flask} - \overline{CO_{bg,flask}}]_i}{CO_{2ff,flask\_i}}.$$  (6)"

Fig. 1: Change "white" -> "black" in the caption.

Fixed in Figure 1 caption, thank you.

p. 10, l. 281ff: From Fig. 2b, it appears that the winter 2017 FT samples are not "slightly higher" than the 3-month averaged GGGRN measurements (only the fall and spring samples seem to be higher). You are correct. This sentence in L314-319 has been adjusted to: "For fall and spring seasons, ACT FT $\Delta^{14}CO_2$ is slightly higher than that in the GGGRN sites, including the upwind, high-altitude (3000-4000 MASL) Niwot Ridge, CO (NWR) site. During ACT, FT samples are frequently obtained from higher altitudes than the GGGRN $\Delta^{14}CO_2$ paired flask samples with fixed collection altitudes, resulting in the potential for ACT FT air to originate from different latitudes or altitudes where influence from cosmogenically-produced $^{14}CO_2$ could explain deviations of ~ 2‰ above GGGRN $\Delta^{14}CO_2$ observations (Turnbull et al., 2009)." We discuss winter data after this, separately, as these are not higher than GGGRN measurements.

Fig. 2: Does the shaded area of the 3-month averaged GGGRN data indicate the 1-sigma standard deviation?

Yes, this is correct. Figure 2 caption now reads b): Daily mean ACT FT $\Delta^{14}CO_2$ background measurements ($\pm$ $1\sigma$ standard deviation shading).

p. 11, l. 294-295: One could add here that negative $CO_{2ff}$ values could also be explained by an inappropriate $\Delta^{14}CO_2$ background or by underestimated nuclear/bio masking of $CO_{2ff}$.

Correct. L326 has been changed to: "ACT observed $CO_{2ff}$ ranges from -0.8 to 15.5 ppm, noting that negative $CO_{2ff}$ values are not physically realistic but can occur due to the measurement uncertainty of $\Delta^{14}CO_2$, by using an inappropriate $\Delta^{14}CO_2$ background value, or by underestimating the $CO_{2corr}$ term in Eq. 4"

p. 11, l. 295ff: Have those studies also used the FT as a $\Delta^{14}CO_2$ background to calculate the $CO_{2ff}$ concentrations?

Graven et al., (2009), Turnbull et al. (2011), and Miller et al. (2012) all use FT data to define $\Delta^{14}CO_2$ background, though there do exist small differences in how $\Delta^{14}CO_2$ data are filtered in these studies for high $\Delta^{14}CO_2$. While the work here takes an average $\Delta^{14}CO_2$ FT value, others look at the lowest values measured in the FT (Turnbull et al., 2011) or filter for anomalously high $\Delta^{14}CO_2$ measured in the FT (Graven et al., 2009). Miller et al., (2020) utilize nighttime high altitude samples representative of clean FT air as a $\Delta^{14}CO_2$ background.

Overall, we use FT as a background for this study because it is not significantly different than other, traditional background estimates (when compared in Figure 2), and these data are available on a daily basis.

p. 13, l. 327: Can part of the weak correlations between $CO_{2tot}$ and $CO_{2ff}$ be explained by the small $CO_{2ff}$ signals?

Absolutely, and we try to reiterate this many times throughout the manuscript. We've changed L356-360 to: "Somewhat strong relationships between $CO_{2tot}$ and $CO_{2bio}$ were found during the March winter 2017 campaign with a regression slope close to 1 but relatively weak correlation ($R^2 \sim 0.3$), which could partially be explained by small $CO_{2ff}$ signals, but also indicate that observed $CO_2$ had a non-negligible influence from biogenic $CO_2$ exchange as was also found in previous studies (Potosnak et al., 1999; Turnbull et al., 2006; Miller et al., 2012; Baier et al., 2020)."

p. 13, l. 330: The spring campaign took place in May, not in March-April, right?

Yes, this is correct – thank you for catching this error. Fixed in L365.

p. 14, l. 332: The fraction of $CO_{2bio}<0$ is similar in fall and winter (8% and 9%, respectively). Could the $CO_{2bio}<0$ data in winter be explained by the fact that the winter campaign took place in March when biosphere is already starting to be active or is it due to observational uncertainties (i.e. is the small fraction of $CO_{2bio}<0$ significant in winter)?

We include error bars in Figures 4 and 5 to indicate the significance of $CO_{2bio}<0$ during winter, but we also add in L356-360 (see comment above) that the winter campaign occurs in March and talk about how this could mean that $CO_{2bio}$ is non-negligible at times. A bit of both could be occurring.

p. 14, l. 341-342: Do you mean: "when combining all seasonal ACT data except fall"?

Yes, this has been clarified in L371: " In Figure 5, we observe a strong positive correlation with $CO_{2bio}$ and OCS in Figure 5 and when combining all seasonal ACT data except fall where $CO_{2bio}$ is either negative or weakly positive (see Fig. S1a of supplementary material), consistent with photosynthesis explaining negative $CO_{2bio}$ values."

Fig. 6: Maybe you could briefly mention in the caption of Fig. 6 that you have changed the ordering of the seasons (compared to the previous plots in Fig. 4 and 5).

Done. Fig. 6 caption now states: "Note that, unlike Figures 4 and 5, flight data is shown in chronological order."

p. 16, l. 377ff: Another possibility for the large variability in $R_{CO}$ are measurement uncertainties. In Maier et al. (2024b, see the appendix A1 of the study), we have seen that the large relative uncertainty of small $CO_{2ff}$ signals can lead to spuriously high $R_{CO}$, even in the absence of natural CO sources:
Maier, F., Levin, I., Conil, S., Gachkivskyi, M., Denier van der Gon, H., and Hammer, S.:
Uncertainty in continuous $\Delta CO$-based $\Delta ffCO_2$ estimates derived from $^{14}C$ flask and bottom-up $\Delta CO / \Delta ffCO_2$ ratios, Atmos. Chem. Phys., 24, 8205–8223, https://doi.org/10.5194/acp-24- 8205-2024, 2024b.

Yes, we agree that low $CO_{2ff}$ signals (and their relatively large associated errors) can create spuriously high $R_{CO}$. As the ACT $CO_{2ff}$ had low signal:noise – mentioned many times in the manuscript and in a general comment above – it is possible that $R_{CO}$ variability and magnitudes could be affected by this. L411-422 discuss several possibilities as to why $R_{CO}$ variability could be high, but then states that it is most likely that low $CO_{2ff}$ is what is causing this.

Fig. 8: Is the distribution of the model-data differences in hourly resolution?

No, the number of bins in the distribution were reduced for visibility. Both Figure 7 and Figure 8 have now been adjusted, with the graphing function automatically choosing the number of bins appropriate to show the range of values and the shape of the underlying distributions in each figure.

p. 20, l. 414: Is the 20% uncertainty in the fossil fuel $CO_2$ fluxes also appropriate for the temporal scale of individual hours of the day? I'm wondering if an inappropriate diurnal profile in the $CO_{2ff}$ fluxes could contribute to the observed model-data differences? For a better interpretation of the results, it would be helpful to show the time axis in Fig. 11 in local time instead of UTC, or to indicate in the caption of Fig. 11 what 16:30 UTC is in local time.

- It is possible that re-gridding of emissions datasets within PSUWRF could create higher uncertainties at hourly, urban timescales. We've added in L455 "Current uncertainties of $CO_2$ fossil fuel fluxes, based on differences among various emissions products, are less than or equal to 20% at the regional scale (Gurney et al., 2020), and this is corroborated in the PSUWRF model, where Feng et al., (2019a) note that the uncertainty in fossil fuel fluxes on a daily average time scale is approximately 20%. This uncertainty may increase at the hourly timescale and certainly within smaller, urban-scale domains (Gately and Hutyra, 2017)."
- We also ran two different emissions datasets (ODIAC and Miller) independently in PSUWRF with the hypothesis that, if the ensemble of these simulations is narrow for $CO_{2ff}$, then we can have more confidence that transport is erroneous rather than fluxes. However, additional simulations that incorporate an ensemble of both emissions and transport datasets are needed to verify this hypothesis and we've noted this in L475-480.
- The time axis in Figure 11 has been changed to local time for the reader. Similarly, the time axis for Figure 10 has also been changed to local time – thank you for noting this potential for improvement.

p. 20, l. 415ff: Could a slight misalignment in the modelled wind direction be the reason for the temporal shift in the maximum of modelled $CO_{2ff}$ in Washington?

We have restructured L456-493 a bit to better clarify the steps we've taken to investigate the reason for the Washington, D.C. plume mismatch between PSUWRF and observations. We first investigated modelled versus measured meteorology along the flight track (wind directions and speeds, boundary layer depth), which all

compared well. This result leads us to think that it is unlikely that a misalignment in the modelled wind direction for Washington, D.C. could result in the temporal shift of modelled $CO_{2ff}$.

We have added additional work that would need to be completed to verify our hypothesis that vertical transport parameterizations are erroneous earlier in the day, leading to improper "venting" or accumulation of $CO_{2ff}$ later in the day as discussed in the comment above.

Technical corrections:

p. 2, l. 43: Delete ", however" or connect it with the next sentence.

Deleted.

p. 8, l. 225: "Maier et al., 2023" -> "Maier et al., 2024"

Fixed.

p. 10, l. 288: Delete „or"?

Deleted.

p. 12, l. 314: "four years" -> "three years"?

Done.

p. 16, l. 381: „Spring" -> „spring"

Fixed.

Fig. 11: „Psuedo" -> „Pseudo" (in the caption)

Fixed.

p. 20, l. 435: "Maier et al., 2023" -> "Maier et al., 2024"

Fixed.

**Reviewer 2**

This paper describes a set of 14CO2 measurements along with CO2 and other trace gases, made from aircraft campaigns in the Mid-Atlantic US region. It examines the relationships between the various measured species to evaluate the various CO2 sources and sinks. It also compares the results with modelled CO2 predictions.

The dataset is good, with careful measurements, and the analysis is good. The paper could be published with minor revisions as noted later, but it feels like a bit of a missed opportunity - the paper demonstrates how 14CO2 measurements *could* be used, but doesn't go so far as to draw any strong conclusions from this dataset. Perhaps this is because this is a large dataset, and exploring all the different facets in detail is probably too much for one paper. I would like to see further analysis on several subtopics, as described below. It seems that there might be a way to do this by adding a few pointers to this paper that allow follow up papers that go into more detail on these topics (whereas as written currently, it might be difficult to write those additional papers without them seeming to be repetition).

We thank the reviewer for taking the time to conduct a thorough review and for their insightful comments and suggestions for improving this manuscript. Please see author responses below in blue that address these comments.

Areas that should be expanded here or in future papers:

1. To my knowledge, this is the first time 14CO2 and OCS measurements have been analysed side by side in this way. In fall, the 14CO2 and CO2 measurements imply a net biogenic CO2 source, but the OCS measurements indicate that significant CO2 drawdown may still be occurring. This is an initially surprising result that could have important implications for biogenic CO2 and potentially could contribute to biogenic model development.

Thank you for this comment. Given current OCS literature that we cite below, we do not believe that this result is as surprising. Please see comment below.

2. The use of CO to develop CO2ff'. There are a few papers now that have used this method, but it is still under development, with little challenges coming up on each new environment in which it is used. In this case, the substantial variability in RCO, particularly in fall and spring, is curious. While the high RCO values in July might reasonably be explained by VOC production of CO, it seems surprising that VOC production would be important in October and May. This variability should be explored, to understand what might be driving it (biomass burning?), and to evaluate how the variability impacts the CO2ff' calculations, and the uncertainty that it might induce. This could be expanded in a subsequent paper, but some of the issues do need to be addressed in this paper (see specific comments).

Please see comments above and below addressing this. In general, we know that the low signal to noise ratio of the ACT 14CO2 data plays a large role in the RCO variability; we address potential VOC oxidation to CO and biomass burning ideas below.

3. As already noted on the paper, there is a large and surprising discrepancy between the observed CO2ff' and modelled CO2ff in the case study from July 24th for Washington DC. The only conclusion currently drawn is that this suggests a model bias, perhaps associated with the time of day. This should be explored in detail, comparing the observed and modelled values for a larger suite of the aircraft observations.

We agree that this discrepancy is large and surprising. We also agree that, ideally, this would be explored in detail and perhaps extrapolated to a larger suite of aircraft observations. We do not, at this point, have the resources for this level of attention for this one flight and it would be beyond our scope to bridge to other flight campaigns with

additional observations around cities. We expect that this is an issue with atmospheric transport but could find no obvious cause in the simulations. We wish to document this case and, as you suggest, point out the value in additional study.

Specific comments:

Line 32 also oceans, not just biosphere.

Done. L33 now reads: "Because total $CO_2$ fluxes encompass biogenic, oceanic and anthropogenic processes, and regional spatial scales are ones at which carbon mitigation strategies are generally developed and implemented, it is important that these $CO_2$ component processes be accurately quantified."

Line 45, suggest referencing Ingeborg Levin's seminal 2003 paper here as well.

Done in L47, thank you.

Line 48. "The remaining variability" rather than "any remaining variability".

Fixed to "…the remaining variability…" in L49.

Line 82. OCS is mentioned only briefly in the introduction, yet the results and discussion use it quite strongly. Some discussion of the utility of OCS for quantifying photosynthetic drawdown should be added.

We add the following lines 73-79: "As biogenic $CO_2$ emissions and sinks – even within cities – are non-negligible, and ignoring these signals could potentially bias fossil fuel $CO_2$ emission inventories, $\Delta^{14}CO_2$ measurements play an important role in determining biogenically-driven ($CO_2$ photosynthesis and respiration) emissions or sinks (Levin et al., 2003; Lopez et al., 2013; Turnbull et al., 2015; Miller et al., 2020). One tracer that could aid in the understanding of $CO_2$ uptake processes alongside $\Delta^{14}CO_2$ is carbonyl sulfide (OCS), which is taken up by plants but not respired (Montzka et al., 2007; Campbell et al., 2008)."

Line 90. How many flasks and 14C samples in each campaign, in total, etc?

We add the total amount of $14CO_2$ samples analyzed over ACT in L135: "In total, 380 $^{14}CO_2$ samples were analyzed throughout the ACT campaigns: 87 in winter 2017, 100 in fall 2017, 104 in spring 2018 and 89 in summer 2019."

Line 99 give approximate ABL heights and the altitudes at which the ABL samples were taken.

L107-111 now states: "Airborne atmospheric sampling during individual research flights was focused at altitudes within the ABL (flight altitudes of 330 m above ground level in most cases) where greenhouse gas abundances are strongly influenced by surface fluxes with occasional sampling in the free troposphere (FT, ~2400 to 9000 m above mean sea level (ASL)) to determine chemistry and isotopic composition of background air against which observed ABL enhancements or depletions are assumed to occur."

Line 125. The methodology was first described in Turnbull et al 2007.

L134 has now been changed to also reference this publication where the methodology was first described. Thank you for this correction.

Line 132. I believe the half-life was calculated as 5730 in this paper.

L142 has been fixed to reflect this more exact half-life, thank you.

Line 140 in equation 1, only CO2bio is included in "other", but in equation 2, the additional terms are added. Be consistent.

We have respectfully chosen to leave Equation 1 as-is because the nuclear term, if written as $CO_{2nuc}$ in Equation 1, is significantly (12 orders of magnitude, ~$400 \times 10^{-18}$ ppm) smaller than other terms. We do acknowledge the omission of the nuclear term in L167 and explain why the term is included in $CO_{2corr}$.

Line 144 see recent Maier et al paper for a different presentation of the nuclear correction.

Again, we have chosen to leave Equation 3 and the nuclear correction stated as-is. While it is possible to rewrite the nuclear term in this equation like all of the others, we feel that the presentation of $\Delta_{nuc}CO_{2nuc}$ does not make as much sense physically, being that the magnitude of $CO_{2nuc}$ is so small.

Line 152. See Turnbull et al 2009 about the assumption that delta-photo equals delta-bg. Turnbull, J. C., et al. (2009). "On the use of 14CO2 as a tracer for fossil fuel CO2: quantifying uncertainties using an atmospheric transport model." Journal of Geophysical Research 114, D22302.

We have noted the limits behind this assumption in L164-166: "$\Delta_{photo}$ is assumed equal to $\Delta_{bg}$ because the '$\Delta$' notation accounts for mass-dependent fractionation processes such as those occurring during photosynthesis. Turnbull et al. (2009) note that this assumption is valid in the limit that time and space between background and observation tends to zero."

Line 182. There is an update on graven and gruber 2011, in: Zazzeri, G., et al. (2018). "Global and Regional Emissions of Radiocarbon from Nuclear Power Plants from 1972 to 2016." Radiocarbon 60(4): 1067-1081.

We have noted this new publication in L211-218: "We note that the calculated magnitude of $CO_{2corr}$ attributable to nuclear reactor emissions in the Mid-Atlantic U.S. is 0.25 ± 0.44 ppm $CO_2$ and does not exhibit a seasonal pattern as it is assumed constant annually. Graven and Gruber (2011) have been used at the time that this publication was written, but updates to nuclear reactor fluxes have been published in Zazzeri et al. (2023). The magnitude of $CO_{2corr}$ attributable to the terrestrial biosphere in the Mid-Atlantic U.S. is approximately 0.10 ± 0.04 ppm $CO_{2ff}$ (1σ) during the winter deployment and exhibits a seasonal cycle with a maximum during the summer of 0.70 ± 0.28 ppm $CO_{2ff}$ (1σ), indicating similar seasonal behavior as that calculated by Miller et al. (2012) for the Mid-Atlantic U.S. In total, the average magnitude of $CO_{2corr}$ (0.8 ± 0.6 ppm) calculated here to correct $CO_{2ff}$ in this work is roughly comparable that first described in Turnbull et al. (2006). "

Lines 195-200. Note that these calculated values are similar to the widely used estimates first described by turnbull et al 2006.

Noted in L211-218; please see above.

Line 230. The median method can be problematic when CO2ff is small. Did you consider using regression to determine RCO?

We have, in fact, considered using regression to determine $R_{CO}$. While we considered using regression to determine $R_{CO}$, it was difficult to perform regressions due to the high scatter and low $R^2$ values (ranging from 0.08-0.33) between flask $\Delta CO$ and $CO_{2ff}$ observed during each ACT seasonal campaign – again, due to the airborne sampling strategy. Rather than utilizing weak regressions, and while there are uncertainties associated with both methods for this dataset (as mentioned in Maier et al., (2024a)), we ultimately decided that the median method for this work was the most robust way to calculate $R_{CO}$ values. $R_{CO}$ is calculated for each flight day as opposed to relying on a single value for each ACT seasonal campaign, and this variability is a large source of the overall uncertainty in $CO_{2ff}$'. This is explained in L265-270 as a limitation to this analysis.

Line 235.  Compare to the recent Maier paper that assessed uncertainties in the different CO2ff methods.

Maier, F., et al. (2024). "Uncertainty in continuous ΔCO-based ΔffCO2 estimates derived from 14C flask and bottom-up ΔCO∕ΔffCO2 ratios." Atmospheric Chemistry and Physics 24(14): 8205-8223.

Please see general comment at the top of this document, and the comment above.

Line 274 "this difference"? Not sure what is being referred to here.

L301-303 has been fixed to clarify: "However, using a higher threshold altitude to define background values can also lead to higher uncertainties in air that is considered as a background for local ABL measurements due to the fact that air at or above 4000 m ASL may have experienced longer-range transport."

Lines 280-285 it seems likely that the higher FT 14C values would be due to the 14C gradient in the FT induced by cosmogenic production.  See Turnbull 2009 and others for some detail on the expected magnitude of this gradient.

L316-319 has been modified to read: "During ACT, FT samples are frequently obtained from higher altitudes than the GGGRN $\Delta^{14}CO_2$ paired flask samples with fixed collection altitudes, resulting in the potential for ACT FT air to originate from different latitudes or altitudes where influence from cosmogenically-produced $^{14}CO_2$ could explain deviations of ~ 2‰ above GGGRN $\Delta^{14}CO_2$ observations (Turnbull et al., 2009). "

Lines 315-317.  Please reference this statement.

Line 342 now references Tans et al., 1990.

Figure 5.  Use OCSxs?  Confusing that this is showing the OCS enhancement over background but is labelled just as OCS.

$OCS_{xs}$ is now used in Figure 5 and throughout Section 3.2. Thank you for this suggestion.

Line 345.  While the explanation for the summer, spring and winter relationships is expected, the fall data is initially surprising. It is hard to believe that soil/litter uptake of OCS would be that significant (and if it is, it is an important finding).  This warrants more investigation.  See my general comment earlier.

Referring to your general comment above alongside this one, other studies have seen year-round OCS uptake (some, of the same magnitude as we calculate here) while net respiration occurs for $CO_2$ in the fall and instances where OCS uptake is decoupled from GPP. Therefore, this result is not so surprising, but more so just consistent with existing OCS literature (cited in L382-394). We clarify that the magnitude of $CO_2$ GPP is not large enough to offset respiration here, and have clarified that other OCS uptake processes (e.g. by soil, leaf litter) could also be occurring that are decoupled from GPP. We do note that further studies could use OCS alongside radiocarbon-based $CO_{2bio}$ have reworked L382-394 as:

" The net negative correlation during fall between $CO_{2bio}$ and $OCS_{xs}$ indicates that, while $CO_2$ uptake could still be occurring, the magnitude of this process is not large enough to offset respiration. Similar to ACT, clear differences in the amplitude and phases of OCS and $CO_2$ cycles were first described in Montzka et al. (2007) and in other, previous studies (Kuai et al., 2022, Ma et al., 2023). Furthermore, many OCS studies have found several instances where ecosystem OCS uptake is decoupled from GPP, including nighttime processes (Commane et al., 2015; Kooijmans et al., 2017; Hu et al., 2021), uptake from soil (Whelan et al., 2022) and/or from senescing or decaying fall vegetation (Sun et al., 2016; Rastogi et al., 2018). Further studies could utilize collocated measurements such as OCS and radiocarbon-based $CO_{2bio}$ to evaluate a) the relationship between OCS and $CO_2$ cycles during the transition from net

photosynthesis to net respiration and b) regional model GPP and respiration fluxes. Further, we note that correlations between negative $CO_{2tot}$ and $OCS_{xs}$ weaken in the early fall due to proportionally high fossil fuel emissions, providing insufficient information about GPP. Parazoo et al. (2021) found that models underestimate observed $CO_{2tot}$ during ACT, and have decreased fidelity in reproducing GPP inferred from observations throughout the U.S. Since stronger relationships emerge between $OCS_{xs}$ and $CO_{2bio}$ than $OCS_{xs}$ and $CO_{2tot}$, again, using OCS and radiocarbon-based $CO_{2bio}$ could further inform and constrain these model processes. ”

Lines 375- the substantial variability in RCO needs some investigation. VOC production of CO is exponentially related to temperature (see Vimont et al 2017), so the high RCO values in July can reasonably be explained by VOC production, but it is harder to explain the fall and spring high RCO values. The high variability for those spring and fall days also suggests that another mechanism might be occurring. I am wondering if biomass burning events could explain them? Alternatively, I wonder if using the median method to calculate RCO could be a factor in these values?

Please see comment above re: the median method and why it was chosen.

Vimont, I. J., et al. (2017). "Carbon monoxide isotopic measurements in Indianapolis constrain urban source isotopic signatures and support mobile fossil fuel emissions as the dominant wintertime CO source." Elementa: Science of the Anthropocene 5(63).

We have referenced the Vimont et al. (2017) study, and have indicated that we expect to see highest RCO values in the summer, yet we do see relatively high values in the spring. DiGangi et al. (2021) have investigated biomass burning influence on ACT flight observations and found negligible influence of this process, especially in the Mid-Atlantic region. We note this in L411-422:

“Figure 6 indicates that the median $R_{CO}$ calculated for all ACT missions (10.03 ± 8.15 $\Delta$CO per ppm $CO_{2ff}$; 68% CI) is slightly higher than in urban studies (Turnbull et al., 2011a; Miller et al., 2012; Turnbull et al., 2014). Day-to-day or even diurnal $R_{CO}$ is also largely variable, which could be a result of variability in $CO_{2ff}$, or in measured background levels (which are represented by a single daily value). Spatial differences in VOC oxidation could play a minor role in influencing the variability of $R_{CO}$. CO additions from VOCs throughout the ACT Mid-Atlantic region will render $R_{CO}$ higher. However, despite the expectation that greater oxidation of VOCs will produce more CO during the summer (Vimont et al., 2017), we see the highest average $R_{CO}$ values during spring. $R_{CO}$ variability in both the fall and spring campaigns suggest that there could be other influential mechanisms occurring. However, DiGangi et al. (2021) found that biomass burning influence on air sampled during ACT were negligible in the Mid-Atlantic region, ruling out the possibility that biomass burning events contribute to the high variability of $R_{CO}$ calculated here. In general, it is more likely that low $CO_{2ff}$ signals with high relative uncertainties are creating abnormally high $R_{CO}$ values using this median method in this work, which has also been found and discussed in great detail in Maier et al., (2024b).”

See also the recent Maier paper that investigates the uncertainty in RCO and calculated CO2ff.

Reviewer 1 also comments on this – see addition to text in L411-422above.

Figure 9. Is there a better way to present these figures? I found them hard to look at with the large, overlapping symbols.

The goal with this figure was to present the model-observation comparisons geographically to show seasonally what Figure 8 depicts in a histogram. However, we agree that the number of points overlapping in this plot make visualization difficult. To help improve the presentation of this figure, we have replaced Figure 9 with a more sophisticated bubble chart with better sizing and transparency of points. We think that this improves over the previous rendition of Figure 9.

Lines 414-430. This paragraph is hard to follow, suggest revising for clarity. Also, see my general comment above.

This paragraph (L456-493) has been revised to indicate several steps that we took to validate the model meteorological variables, and investigate potential differences in flux products used. We do provide a hypothesis that model transport rather than fluxes are erroneous. However, referencing the general comment above, it is beyond the scope of this observation-based work to more thoroughly investigate the mismatch in the Washington, D.C. plume. Instead, we highlight the importance of documenting this case and outlining additional thoughts for modeling research areas that could help prove or disprove this hypothesis and improve future model comparisons to $CO_{2ff}$.

L475-479 state: "Therefore, while our current results suggest that model transport, rather than fluxes are erroneous on this particular flight, the results from this case study are important to document for future model-observation comparisons. More intensive work beyond the scope of this work would be needed to verify our hypothesis, which could involve model runs with a number of both transport and flux variants to discern model variability with different, realistic ensemble members. Within this type of study, observed and modeled values for additional urban campaign data should also be compared."